# The alignment of companies' sustainability behavior and emissions with global climate targets

Simone Cenci [1] ✉, Matteo Burato [1], Marek Rei[1,2] & Maurizio Zollo[1] ✉

Climate actions by the private sector are crucial to cutting global emissions and meeting the climate targets set by the Paris Agreement. However, despite an increasing number of climate pledges, the emissions pathways of most companies are still misaligned with the Paris targets. To identify the causes of this discrepancy between effort and outcome, we developed a systematic approach, based on extensive analyses of textual data, to track the actions implemented by major public corporations to reduce their emissions. Our findings suggest that the misalignment between companies' climate goals, actions, and outcomes is due to a widespread over-investment in risk mitigation actions as opposed to innovation and cooperation activities to foster energy goals. Overall, we provide a systematic framework to track companies' climate actions. Our approach can be used by investors and policymakers to redirect capital towards its most sustainable use and to design behaviourally founded climate policy interventions.

To limit global warming within the goals set by the Paris Agreement countries have put forward emission targets, but meeting these targets depends on the actions of non-state actors, most notably corporations[1,2]. Indeed, a significant component of global greenhouse gas (GHG) emissions can be directly associated with business activities, from resource extraction and industrial production to transportation and land use[3]. Therefore, changes in corporate behaviour are crucial to reducing the impact of human activities on long-term climate dynamics[1,4].

Since the adoption of the Paris Agreement, a large number of publicly traded corporations have pledged to lower their emissions to a level compatible with the temperature targets of the Agreement[5–8], and a significant amount of capital has been allocated to support the pledges[3,9]. However, as of 2020, out of ~13600 large public corporations, only ~19% (~2500) have emissions pathways aligned with these targets (Supplementary Fig. S3). Therefore, despite a long series of commitments and the unprecedented flow of resources towards supposedly environmentally sustainable funds and companies[10], the private sector is struggling in delivering the transition towards a sustainable economy[5].

Identifying, explaining and addressing the root causes of this failure requires, as a first stepping stone, an understanding of what companies are doing to lower their emissions to a level compatible with the Paris targets and of what type of actions are effective in lowering the impact of business operations. Such an understanding is crucial for (1) business leaders to learn what to focus on in their effort to lower emissions, (2) market participants to allocate capital towards its most sustainable use, and (3) policymakers to devise effective intervention strategies to curb emissions. However, the task is complicated due to a lack of a systematic reporting framework for corporate climate actions and spending, which makes monitoring corporate sustainability behaviour (climate actions and goals) and the outcome of the behaviour (GHG emissions and their projections) a particularly cumbersome task[5,11,12].

Previous works that look at companies' contributions to the achievement of climate targets have primarily focused on the analysis of commitments (e.g., whether or not a company has set emission targets and the type of target[5,13,14]) and high-level climate actions (e.g., disclosure of emissions and business costs, and the extent to which climate change responsibilities are delegated to the board or senior

[1]Leonardo Centre on Business for Society, Imperial College Business School, London SW7 2BX, UK. [2]Department of Computing, Imperial College London, London SW7 2BX, UK. ✉e-mail: s.cenci@imperial.ac.uk; m.zollo@imperial.ac.uk

management[13]). Other works looked at the management practices in further detail by analysing standardised datasets such as, for example, the climate actions self-reported to the Carbon Disclosure Project (CDP)[15,16].

Here we take a different approach. Specifically, we develop a systematic framework using natural language processing approaches to identify and characterise company actions to reduce their emissions. We focus on implemented actions (not company-level commitments), we look at a broad spectrum of actions (not only those reported to CDP), we analyse the goals of those actions (the whole sustainability behaviour), and we focus on a large number of companies (~4000), countries (51), sectors (11) and years (10).

To collect information about companies' climate actions and goals, we use information disclosed in sustainability reports: annual reports that describe the activities a corporation has undertaken during a given fiscal year to address societal problems, from lowering emissions to reducing inequality in their management, workforce and local communities. Several studies have looked at the information content of sustainability reports (see ref. 17 for a recent analysis and ref. 11 for a comprehensive review). However, the difficulty in collecting historical reports for a sufficiently large number of companies, the lack of clear reporting standards and the resulting lack of comparability and quantifiability of the information content of sustainability reports are major limiting factors for their analysis[18]. Indeed, we still need a database that systematically maps the unstructured information contained in the text of the reports into objective, quantitative and material information about corporate climate actions and goals. Here we build such a database using natural language processing approaches to search, identify and classify climate actions and goals for the major publicly listed companies around the globe.

Our process is organised as follows (see section Behavioural dataset in the Methods for further detail): first, we develop an extensive training set by manually annotating 500 sustainability reports to identify corporate environmental actions or initiative (we use the terms interchangeably). We define an environmental action as an activity implemented by a company (e.g., development of new products, donation and funding, changes in operating processes) to meet a specific sustainability goal, which we classify based on the most closely related Sustainable Development Goal (SDG). For example, an investment in research and development (activity) to increase the energy efficiency of a particular production process would be classified as an activity that targets SDG 12, see Supplementary Section C in the Supplementary Information for a few examples of extracted and classified initiatives. Then, we train two large language models to identify those initiatives and classify them based on the type of activity and the most closely related SDG. Subsequently, we collect the sustainability reports of a large sample of publicly traded companies by systematically crawling them, purchasing them from third parties, and manually searching for them. Next, we run the trained algorithms on all available reports, and for each report (i.e., company-year observation), we count the total number of initiatives classified by activity and most closely related SDG (see Supplementary Fig. S1 for a schematic representation of our process). Finally, we extract all those initiatives directly related to lowering GHG emissions (see Methods, section Behavioural dataset). The final dataset tracks the climate actions of every publicly traded company in our sample through the observation period 2010-2020.

In the following, we will refer to a particular combination of activity/SDG or climate action - as a climate-related sustainability behaviour, or simply sustainability behaviour. This nomenclature follows classic definitions of behaviour, which can be defined as the combination of actions undertaken by an agent (e.g., a company) to achieve a particular goal[19]. The choice of focusing on SDGs as a goal-setting framework is motivated by the finding in the latest assessment report from the Intergovernmental Panel on Climate Change (IPCC),

which illustrates that meeting the targets of the Paris Agreement requires effort from the private sector to meet all the UN SDGs[20]. Therefore, as governments and international institutions face growing pressure to realise the SDGs and to incorporate them within their nationally determined contributions (NDC), companies will be forced to align their behaviours with these targets[21,22] and to report their initiatives within this framework[23].

The objective of this study is twofold. First, we develop a methodology to study corporate sustainability behaviour using natural language processes approaches, and we use our dataset to provide an in-depth, large-scale analysis of the distribution and the temporal evolution of sustainability behaviour with a specific focus on the effort of companies in the hard-to-abate and energy-intensive sectors to lower emissions. Second, as an application of our framework, we show that there are significant behavioural differences between companies that are able to lower their emissions to a level compatible with global climate targets and companies that fail to achieve these goals. To this end, we analyse our dataset in conjunction with other datasets purchased from third party data providers. Finally, we discuss the business and policy implications of our findings. Overall, our approach contributes to the ongoing effort of monitoring companies' actions to align their operations with the United Nations 2030 Agenda and the goal set by the Paris Agreement.

## Results and discussion

In the following sections we present our dataset and an overview of the sustainability behaviour of a large population of publicly listed companies. Then we focus on companies in the hard-to-abate and energy-intensive sectors and we compare the sustainability behaviours of companies successful in lowering their emissions to a level compatible with the targets set by the Paris Agreement versus the behaviour of those that are misaligned with the climate targets.

### A systematic categorisation of sustainability initiatives

Our population comprises 4191 publicly traded companies listed in major exchanges worldwide with a homogeneous distribution across both sectors and geographies (see Supplementary Fig. S4 panels a, b in the Supplementary Information). The inclusion criteria include the availability of accounting and emission data and whether or not a company has published a sustainability report during the observation period 2010–2020. Importantly, our sample covers ~70% of global (public) market capitalisation and invested capital, ~80% of the direct and first-tier indirect emissions available for public corporations, and ~50% of global emissions (see Supplementary Fig. S4 in the Supplementary Information and Methods). Supplementary Table ST4 provides summary statistics of the variables in our sample.

In the Methods section and the Supplementary Information we provide a detailed description of our data-collection process. Briefly, for each company in our sample, we download or purchase their yearly sustainability reports (when available). Then, we train two language models (BERT and RoBERTa) on a large manually annotated training set to (1) identify sustainability initiatives and (2) categorise the initiatives based on the type of activity undertaken by the company (e.g., a research and development investment, the deployment of new products, training of employees) and the most closely related SDG that the activity is meant to target (i.e., the objective of the action). To reduce the risk of double counting initiatives we perform our classification task using the text of the initiatives as well as its context (the preceding and subsequent text) as explained in Methods. Whilst in our process we collect data on all SDGs, here we focus exclusively on environmental SDGs (6,7,11,12,13,14,15). In the Supplementary Section B. we provide a full description of our taxonomy of activities. In Supplementary Section C. we provide some examples of the initiatives and their categorisation.

Each environmental SDG comprises multiple targets, but most of these targets are not related to reducing GHG emissions. For example, SDG 12 includes targets related to reducing food waste (target 3), general waste (target 5) and increasing transparency in reporting (target 6). Because here we are interested in the initiatives implemented in order to reduce GHG emissions, we extract from the total number of initiatives only those related to this particular issue (see Methods). Figure 1 panel a shows how the activities are distributed across SDGs in our sample. The figure shows the Sankey diagram of a matrix where each row is an activity and each column an SDG. Therefore, each cell in the matrix represents the total number of initiatives detected in the report. We refer to this matrix as our

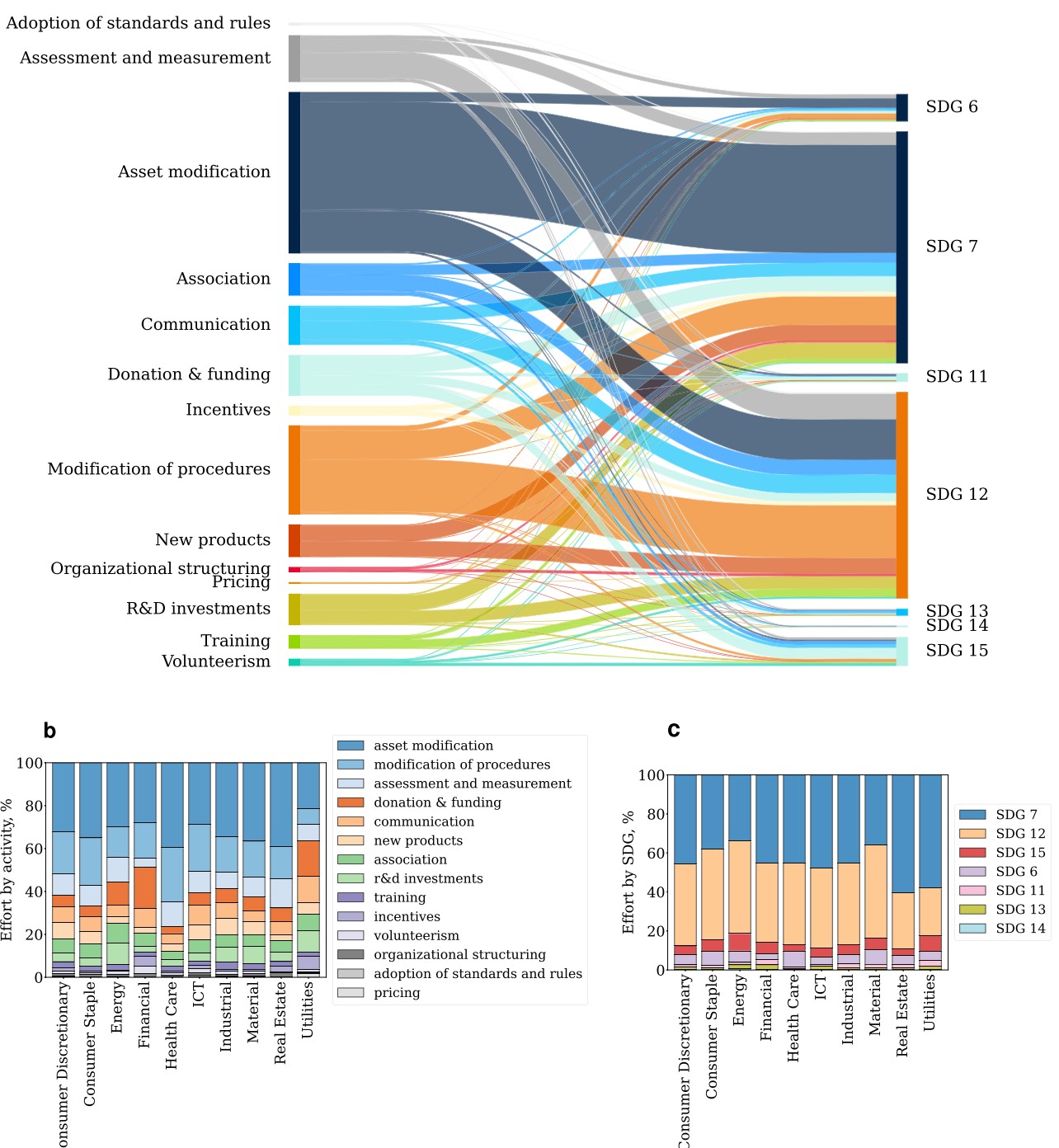

**Fig. 1 | Sustainability behaviour.** Panel **a** provides a summary view of the distribution of the sustainability initiatives. The panel shows the Sankey diagram of the behavioural matrix (Supplementary Fig. S7). Each line in the diagram represents an activity implemented to meet one of the environment-related SDGs. The SDGs on the right-hand side are coloured based on the most prevalent actions. The thickness of each activity is proportional to the relative representation of the activity in the population. The panel illustrates that most activities are changes of assets in place and modification of procedures implemented to align the company with SDGs 7 and 12. Panel **b** and **c** show the sustainability effort by sector, i.e., the number of occurrences of a particular activity (or SDG) divided by the total number of activities (or SDGs) in the sector. Overall, the figure provides an overview of the activities that companies implement to lower their emissions.

behavioural matrix (see Supplementary Fig. S7 in the Supplementary Information). In our framework, a sustainability behaviour is a specific allocation of sustainability effort, i.e., a specific configuration of the behavioural matrix.

Figure 1 panel a shows a large degree of heterogeneity in both activities and SDG targets. Specifically, we have found that most of the activities are asset modifications and modification of procedures intended to meet sustainability goals related to SDG 12 (responsible consumption and production) and SDG 7 (affordable and clean energy). Examples of these initiatives can be found in Supplementary Table ST1 in the Supplementary Information. SDG 13 (climate action) is poorly represented in our sample. At first, this result could be surprising as SDG 13 is the most relevant goal for tackling climate change. However, it is important to notice that the targets of SDG 13 are related mostly to country-level initiatives (e.g., "Integrate climate change measures into national policies, strategies and planning", "Strengthen resilience and adaptive capacity to climate-related hazards and natural disasters in all countries").

When interpreting the Sankey diagram of the behavioural matrix in Fig. 1 it is important to bear in mind two important limitations. First, all the initiatives reported in the panel are accounted for independently of their complexity. Therefore, it is not surprising that activities such as donation & funding (which are easily implemented) are considerably more common than, for example, investments in research and development (which require a substantial effort). Second, the total number of initiatives in the population is likely considerably larger than the one reported here. This is because we impose a strict definition of what an initiative is in order to only include in the analysis initiatives that require a substantial effort (see Methods and Supplementary Section A for a more in-depth discussion).

Figure 1 panel b and c show the distribution of the SDGs and activities, respectively, across sectors. The y-axis in the panels shows the number of occurrences of a particular activity divided by the total number of activities in the sectors. Overall, we have found a strong homogeneity in the SDG behaviour and considerable heterogeneity in the activities. For example, companies in the Financial sector implement a large number of donation & funding initiatives and only a limited number of research & development (R&D) initiatives. On the other hand, companies in the Energy and Material sectors are those with the largest effort in R&D. It is important to notice that some of the differences in the number and relative frequency of activities across sectors are likely due to the nature of the assets of the companies (the proportion of tangible versus intangible assets and the energy needs for production) which require different approaches to decarbonisation and sustainability in general.

## The sustainability behaviour of companies in high-emitting sectors

In order to appropriately compare the sustainability behaviour across our sample it is important to focus on companies with comparable business needs (this follows from the expected relationship between required behaviour for decarbonisation and the nature of companies' assets). Therefore, in this and the following sections, we restrict our analysis to four sectors: Energy, Material, Industrial and Utilities, as defined by the Global Industry Classification Standard (GICS). In contrast to sectors such as Financial and ICT where revenue strongly depend on the value of intangibles (e.g., patents), the business models of companies within these sectors are comparable in that production and revenue strongly depend on tangible assets (~40% of total assets in these four sectors are tangible assets, e.g. plant, versus ~20% in the other sectors) as well as the price of fossil fuels. The main reason why we focus on companies in these sectors is that their actions are crucial to meet country-level nationally determined contributions given the sheer size of their emissions compared to those of companies in less energy-intensive sectors. Indeed, companies in these sectors account

for ~90% of the emissions in our population (see Supplementary Fig. S8)

In these sectors our sample comprises 1951 companies, 9330 reports and 26944 climate related sustainability initiatives (see Supplementary Fig. S9) between 2010-2020. Supplementary Table ST5 shows the summary statistics of companies in this subsample. The average number of initiatives per report as well as the total number of initiatives (red) and reports (blue) per year, is shown in Fig. 2 panel a. The panel shows that, on average, we observe a very limited number of GHG reduction-related initiatives per report (from 2 to 10, depending on the sector and year). Importantly, the figure also shows that the total number of initiatives dropped considerably in 2015 and 2016. This trend is particularly evident in the Energy and Utility sector (see Supplementary Fig. S10 in the Supplementary Information).

Interestingly, Fig. 2 panel b shows that, while the average number of initiatives per report is small, there are companies in the sample with a large number of initiatives. Specifically, the y-axis shows the fraction of companies with less than n% of the total number of initiatives in the sector mentioned across all companies' reports (x-axis). The diagonal line represents a hypothetical uniform distribution. The larger the deviation from the diagonal the more skewed the distribution. For example, in the Industrial sector (blue line) ~85% of companies take on less than 20% of the total number of initiatives. Overall the panels show that the distributions of the number of initiatives are (1) substantially skewed and (2) ssubstantially different across sectors. Supplementary Fig. S11 in the Supplementary Information shows that the skewness of the distributions is also a function of size, with the top 0.1% largest companies taking as many as 18 times the median number of initiatives of the full population.

Figure 2 panel c shows the trend in the number of initiatives after grouping them in macro-categories based on the type of activities following a similar taxonomy as that proposed in ref. 24 (see Methods). The panel shows that there is a strong negative time-series correlation between innovation activities (red, e.g., R&D investment, new products) and activities aimed at managing existing assets and risks (dark orange, e.g., asset modification). On the other hand, innovation activities are positively correlated with reputation and stakeholder engagement activities (blue, e.g., communication, donations & funding). Later in this section we will show that these differences among macro-categories are particularly relevant in differentiating companies' behaviour.

## Using sustainability behaviour to explain companies' alignment with climate targets

One of the main objectives of this study is to use the sustainability behaviour dataset to explain the behavioural differences between companies that are able to lower their emissions to a level compatible with global climate targets and companies that struggle to achieve these goals.

Data on companies' alignment with climate targets are from Trucost, which is the leading provider of corporate emissions and environmental impact data[25]. Following Trucost methodologies, we consider a company to be aligned with a climate target if its projected emission pathway as of a given year (2018, 2019, and 2020) is below the required pathway to limit global warming below 2 °C. In the Supplementary Information we test the robustness of our results to a more stringent target. Emission pathways are computed by Trucost using the Sectoral Decarbonisation Approach (SDA) and the Greenhouse gas emissions per unit of value added (GEVA) approach, see Climate targets. In the main analysis we focus on alignment values calculated as of 2019, since in 2020 the COVID-19 pandemic and global lockdown have caused a substantial exogenous shock to energy companies. However, in the Supplementary information we use data from 2018 and 2020 to test the robustness of our main results.

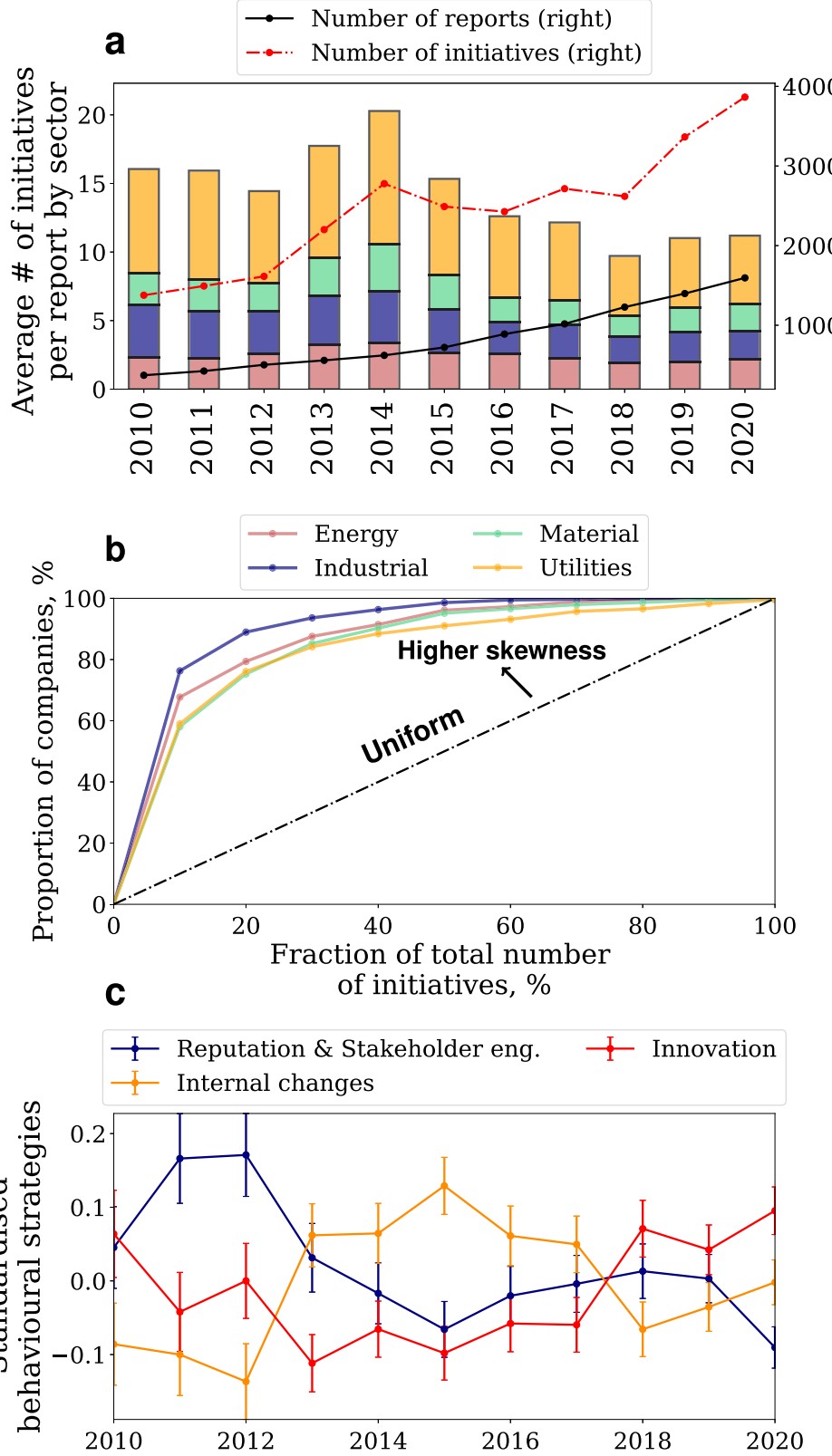

**Fig. 2 | Sustainability initiatives in the energy-intensive sectors.** Panel **a** shows the average number of initiatives per report (bar plot, left axis and colour legend on top of panel), the total number of reports (black line, right axis) and the total number of initiatives (red line, right axis). Panel **b** shows the skewness of the distribution of the initiatives compared to a uniform distribution (black diagonal line). Panel **c** ($N = 9330$) shows the temporal evolution of three different sustainability strategies (shown on a standardised scale to compare trends). Circles and error bars in panel **c** show means and standard errors of the means, respectively. Overall, the figure shows a significant heterogeneity in the sustainability behaviour of companies in our sample.

In the following analyses we divide our population into two groups: one which comprises companies aligned with the climate target and one that includes the misaligned ones. Figure 3 panel a shows an interesting pattern in the GHG emission dynamics of the two groups. Companies with emission pathways aligned with those compatible with the 2 °C target were, on average, greater polluters at the beginning of the 2010s. Yet they have been able to reduce their emissions substantially throughout the decade. This reduction is not due to a size effect, as shown in Fig. 3 panel b, which shows the dynamic of emission intensity, i.e., GHG emissions over revenue. Importantly, note that here we use the default definition of GHG emissions from Trucost which includes all emissions under the direct control of management (see Methods). Panel a and b report statistics in our sample but we do not have continuous observations for every company in the observation period. The full emission statistics for companies aligned and misaligned with the target, including those for which we do not have continuous behavioural data, are shown in Supplementary Fig. S12. The pattern is qualitatively the same: aligned companies have reduced their emissions proportionally more than misaligned companies through the observation period. Therefore, our sample does not have a particular bias in the emission dynamics.

What have aligned companies done differently to lower their emissions to levels compatible with global climate targets? In the following analyses we will try to answer this question. Firstly, we note that the total number of GHG-reduction initiatives is unrelated to the magnitude of the deviation of the target and the probability of observing the alignment. Specifically, Fig. 3 panel c shows the distribution of the deviation from the target as a function of the quartile of initiatives. Negative deviation values correspond to greater alignment, while positive values correspond to greater misalignment.

Overall, the panel shows that companies that take on more initiatives are on average, and distributionally, more misaligned. However, this effect is entirely driven by scale factors. To illustrate this point, in panel d we estimate a model to measure the association between the total number of initiatives and (1) the magnitude of deviation (top table), (2) the probability of being aligned with the target (bottom table).

We describe the model in detail in the Methods, section Sustainability initiatives and alignment with climate targets. Briefly, we assume that the number of initiatives depends mainly on (1) the available capital to finance the initiatives, which can in turn be divided into capital raised in capital markets and revenue, and (2) the nature of the assets of the company (whether revenue is generated from tangible or intangible assets). We also control for historical emissions, which is a key factor in the estimation of the projections, and therefore an important possible confounder of the effect we are after. Then, we control for country and sector fixed effects to account for differences in regulatory frameworks and technological basis. Next, we account for the voluntary nature of the disclosure of sustainability reports with the Heckman correction[26], and we also include fixed effects for reports that follow GRI standards and fixed effects for reports that are subject to audit processes. Finally, we include an indicator variable to distinguish observations with alignment calculated from self-reported and estimated emissions data. Figure 3 panel d show that after controlling for the total level of emissions (second column top table) the positive, albeit not statistically significant, correlation between the total number of initiatives and alignment become negative, but still not statistically significant. The bottom table in the panel shows that the total number of initiatives is positively related to the probability of alignment, i.e., the greater the number of initiatives the greater the probability of alignment, but the coefficients are not statistically significant.

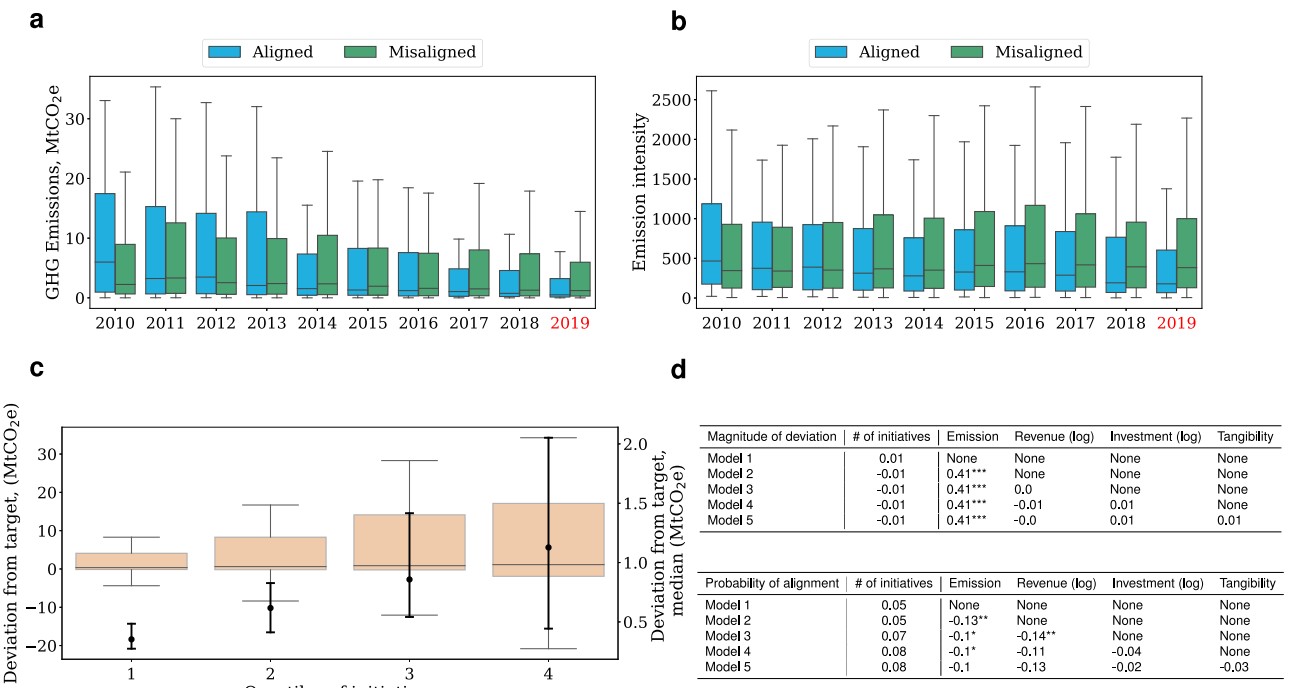

**Fig. 3 | Alignment with climate targets does not depend on the number of initiatives.** Panel **a** and panel **b** show the temporal evolution of the distribution of GHG emissions (**a**) and emission intensity (**b**, emission in tCO$_2$e over revenue) of the companies in the population with aligned (blue, $N = 2255$) and misaligned (green, $N = 4984$) emission pathways as calculated in 2019 (red). Panel **c** ($N = 2184$) shows the distribution of the magnitude of deviation (y-axis) as a function of the quartile of initiatives (x-axis). The lines of the box plots are median lines, and the edges of the boxes are the quartile range: the 25th and 75th percentile. The right y-axis shows the medians of the distributions and their 95% bootstrapped confidence intervals. Panel **d** shows the estimation of the association of the total number of initiatives with the magnitude of the deviation (top) and the probability of alignment (bottom). In each model, we control for fixed effects, source of emission data, self-selectivity, and we adjust standard errors for heteroskedasticity. *, **, *** denote statistical significance at 10%, 5%, and 1%, respectively, calculated from the two-tailed *p*-values for the t-statistics of the parameters. Overall, the figure shows that aligned companies have decreased their emissions through the observation period, but the total number of GHG-reduction initiatives cannot explain the alignment.

Overall, Fig. 3 shows that companies with emission pathways aligned with those compatible with limiting global warming below 2 °C have been able to considerably reduce their emissions. However, the number of sustainability initiatives is unrelated to their capacity to align emissions with the targets of the Paris Agreement, i.e., doing more does not necessarily imply emitting less. In the next section, we show that the relevant explanatory variable for alignment with climate targets is the particular sustainability behaviour that a company implements, i.e. the particular combination of activities and SDGs. In other words, we show that what companies do is more important than how much they do.

## The importance of sustainability behaviours to meet climate targets

In order to identify the relationship between sustainability behaviour and climate targets we perform an ex-post analysis by looking at the differences in the behaviour of companies with emission pathways aligned and misaligned with the target set by the Paris Agreement to lower global warming below 2 °C. For comparison purposes and given the large-scale effects shown in the previous analysis, we first focus on the largest companies in the sample and we fix the target calculation to 2019. Importantly, these companies account for 67% of the sectors' emissions and are therefore a relevant sample to focus on. However, later in the section we extend our analysis to the full population, to different time windows and to the investigation of alignment with a more stringent target. In this group we have data for 379 companies, 119 of which are aligned with the goal of limiting global warming below 2 °C (aligned population), and 260 are not (misaligned population). Note that we have excluded from the analysis companies with less than two years of observations prior to the measurement of the alignment because the model we use to test the significance of the behavioural differences requires historical values of the control factors.

The average number of initiatives in the misaligned and aligned groups are 4.3 and 4.8, respectively, and the difference is not statistically significant (*p*-value > 0.1). On the other hand, the types of initiatives that the two groups undertake are substantially different. Specifically, Fig. 4 panel a and b show the excess effort of companies aligned with the 2 °C target. The excess effort is defined as the difference in the relative incidence of an activity (a) or SDG (b) in the aligned population versus the misaligned population (see Methods). Companies aligned with the target focus more on innovation and stakeholder engagement activities, such R&D investments, association, new products, and communication. On the other hand, companies with emissions pathways misaligned with the targets of the Paris Agreement focus more on the management of existing assets and procedures (e.g., asset modification). We also observe a substantial differential behaviour along the SDG dimensions. Specifically, aligned companies focus more on energy-related goals (SDG 7) than general sustainability objectives (SDG 12).

It is important to notice that the values shown in the panels are the sum of the rows and columns in the full differential behavioural matrix, which is shown in Supplementary Fig. S13 in the Supplementary Information. The matrix highlights important details that are masked in the summary view shown in Fig. 4 panel a and b. For example, the strong negative value of asset modifications is driven by SDG 12, while excess effort for asset modifications aimed at SDG 7 is positive. Similarly, companies exhibit a positive excess effort in research and development investments in SDG 7 and a negative, albeit small, excess effort in R&D for SDG 12. Overall, the matrix illustrates that the behaviours of the two populations differ in a few key activities, but most importantly in the objectives of those activities (i.e., the SDGs).

To assess if the differential behaviour is associated with significant differences in emission alignment we re-evaluate the models shown in Fig. 3 panel d using alternative measures for initiatives. Specifically,

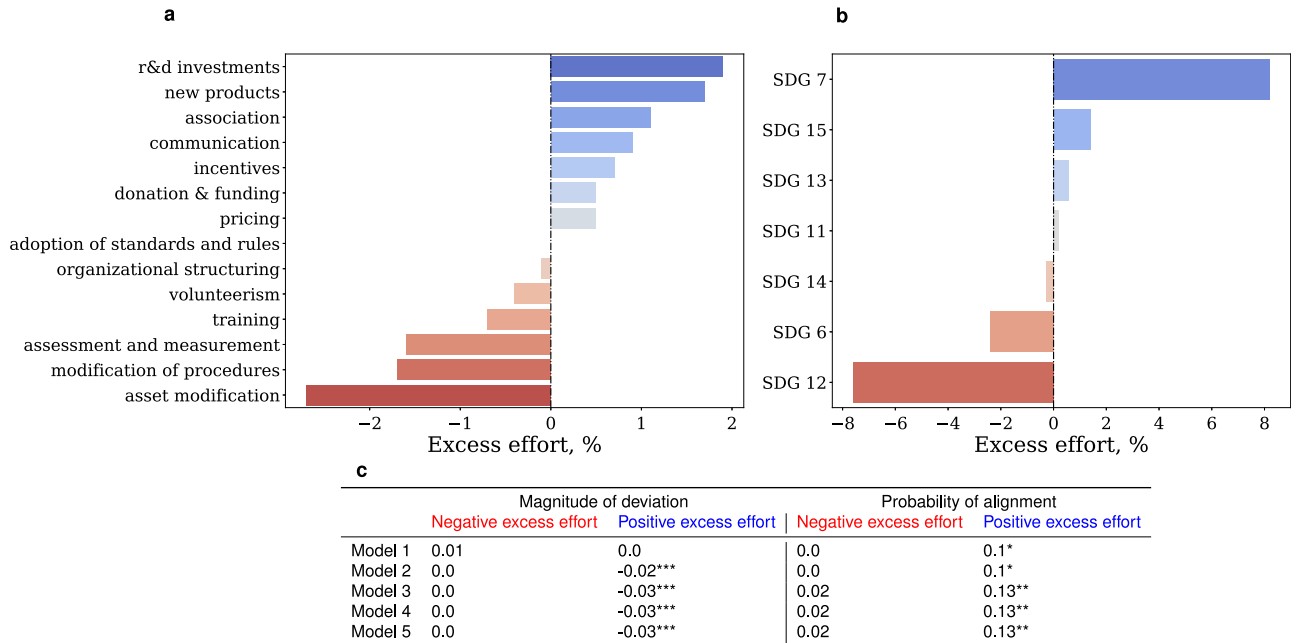

| | Magnitude of deviation | | Probability of alignment | |
| | Negative excess effort | Positive excess effort | Negative excess effort | Positive excess effort |
|---|---|---|---|---|
| Model 1 | 0.01 | 0.0 | 0.0 | 0.1* |
| Model 2 | 0.0 | -0.02*** | 0.0 | 0.1* |
| Model 3 | 0.0 | -0.03*** | 0.02 | 0.13** |
| Model 4 | 0.0 | -0.03*** | 0.02 | 0.13** |
| Model 5 | 0.0 | -0.03*** | 0.02 | 0.13** |

**Fig. 4 | Differential behaviour can explain alignment with climate targets.** Panel **a** and **b** show the excess sustainability effort of companies with emission pathways aligned with the climate targets. The excess effort is the difference in the relative incidence of activities and SDG in the two populations. Blue (red) bars indicate activities and SDGs that are more prevalent in the aligned (misaligned) population. Panel **c** shows the estimation of the association of the total number of initiatives in negative (red) and positive (blue) excess effort with the magnitude of the deviation and the probability of alignment. The model number on the leftmost column in the table corresponds to the incremental addition of control factors as in Fig. 3 panel d. In each model, we control for fixed effects, source of emission data, self-selectivity, and we adjust standard errors for heteroskedasticity. *, **, *** denote statistical significance at 10%, 5%, and 1%, respectively, calculated from the two-tailed *p*-values for the t-statistics of the parameters. Overall, the figure shows that companies with emissions pathways aligned with climate goals focus on behaviours that prioritise innovation and stakeholder engagement activities to realise SDG 7 and that this differential behaviour can explain alignment with climate targets.

instead of regressing the magnitude of deviation and the probability of alignment on the total number of initiatives, we estimate two models independently. In the first model, we use as independent variable all initiatives in negative excess effort across activity types and SDGs (red bars in Fig. 4 panel a and b). In the second model, the independent variable includes all initiatives in positive excess effort across activity types and SDGs (blue bars in Fig. 4 panel a and b). The results of the regressions are shown in Fig. 4 panel c. The table in the panel shows that after accounting for asset characteristics, fixed effects, self-selectivity and the source of emission data (see Methods) initiatives in positive excess effort are associated with greater alignment, both in magnitude and probability. On the other hand initiatives in negative excess effort are unrelated to alignment. Importantly, the results are weaker when we construct the differential behaviour variable by focusing on the activity type or the SDGs independently (Supplementary Table ST7 in the Supplementary Information). This result illustrates the importance of characterising corporate behaviour in terms of both the what (i.e., activity) and the why (goals) of corporate climate actions.

In Fig. 4 we focused on companies in the largest size quartile which is the group that drives most of the emissions in our sample. However, to assess the robustness of our results across the companies in the sample, we re-evaluate the behavioural differences in the full sample and in each size quartile independently. Moreover, we also repeat the analysis for the alignment calculated as 2018 and 2020. Results are shown in Fig. 5. Blue circles in the Figure correspond to

activities and SDGs in positive excess effort (just as the blue bars in Fig. 4 panel a and b). The intensity of the colour is proportional to the deviation from zero, which corresponds to the case of no behavioural differences between the two groups. The full numerical tables are shown in Supplementary Table ST10. Overall, the Figure shows that, while there are some size fluctuations, on average results are robust across different subsamples and time windows. To further confirm the robustness of the regression results, we re-estimate the models of the magnitude of deviation and the probability of alignment in each size quartile and for each alignment estimation year. Results are shown in Supplementary Tables ST9 and are consistent with the previous findings, i.e., initiatives over-represented in the aligned population are negatively and statistically significantly associated with the deviation from the target (greater alignment), while initiatives in negative excess effort are not associated with the deviation. As a further robustness check, we repeat all the previous analyses using as target variable the alignment with a well below 2 °C target. Results are shown in Supplementary Fig. S14, and Supplementary Tables ST11 and ST12.

### Differential behaviour and topics of the disclosures

To further investigate the nature of the behavioural differences shown in Figs. 4 and 5, we run a topic analysis on the text of the initiatives reported by companies in the aligned and misaligned populations (see Topic analysis). The analysis identified seven distinct topics which are shown in Fig. 6. Similarly to the results found in Fig. 4, we have found that innovation investment in the aligned population is mostly focused

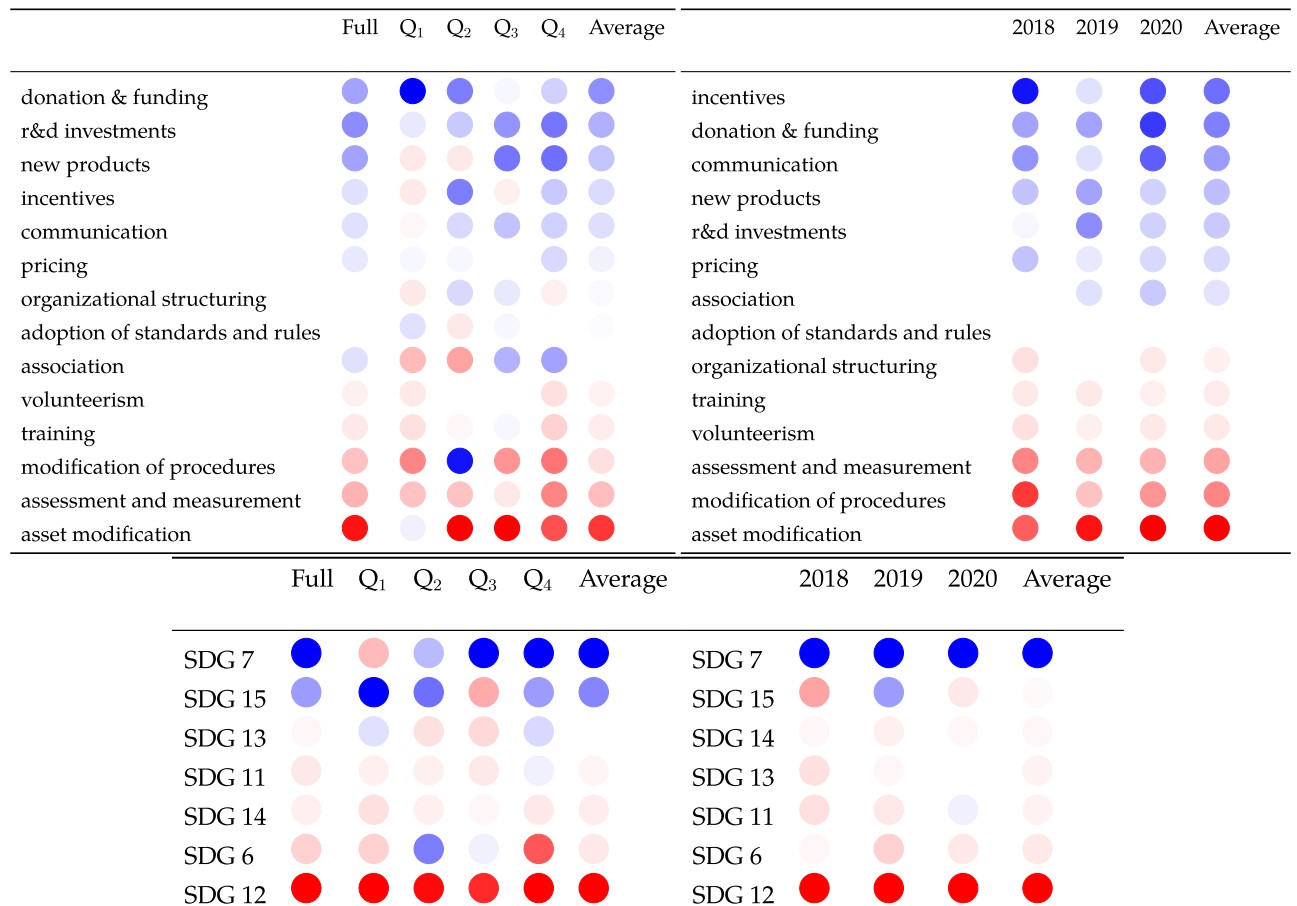

**Fig. 5 | Excess effort across size and estimation windows.** The left tables (top and bottom) show the excess effort of aligned companies across the full population and size quartiles. The right tables (top and bottom) show the excess effort across time. Similarly to Fig. 4 panel c, the blue circles denote actions and SDGs more prevalent in the aligned population, while the red circles denote actions and SDGs more prevalent in the misaligned population. We sort actions and SDGs by the values in the "Average" columns. The intensity of the colours are proportional to the deviation from zero (white). Overall, the tables show consistency of the behavioural differences, particularly when measured across time in the full population.

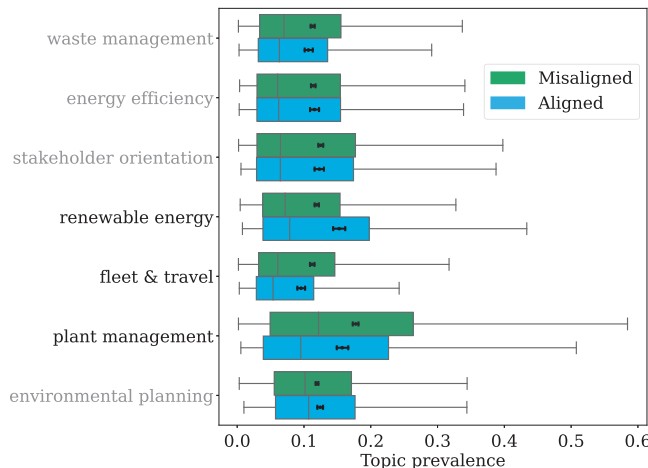

**Fig. 6 | Topic analysis.** The figure shows the historical prevalence of the most relevant topics identified in the initiatives of companies in the aligned (blue, $N = 2255$) and misaligned (green, $N = 4984$) populations. The lines of the box plots are median lines, the full circles are means with their 95% confidence intervals and the edges of the boxes are the quartile range: the 25th and 75th percentile. Overall, the figure illustrates that misaligned companies are more concerned about the management of existing assets (plant and fleet management), while companies with emission pathways aligned with the targets set by the Paris Agreement focus more on topics related to renewable energy sources (which is a topic related to both innovation and SDG 7).

on renewable energy sources, a topic strictly related to SDG 7 (see for example target 7.2 and 7.a), while companies in the misaligned population (green bars) focus more on the management of existing assets (plant and fleet management). Keywords for the topics are shown in Supplementary Table ST13 in the Supplementary Information.

To further analyse the relationship between sustainability behaviour and investment in renewable energy sources in the Supplementary Section C.1 we study the behaviour of a sample of power companies for which we could collect data on the relative composition of energy sources in their energy mix. Supplementary Table ST3 shows that excess effort in the aligned population is associated with a significantly larger presence of renewable sources in the energy mix of the companies. This result further confirms the importance of the topic in the aligned population.

## Main findings and limitations

Emissions from business operations are one of the leading causes of anthropogenic greenhouse gas emissions and global warming[13]. Business leaders of the majority of large public corporations have pledged to align the emissions of their companies with the target set by the Paris Agreement of limiting global warming well below 2 °C[8]. Yet, emissions from many public corporations continue to rise (see Supplementary Fig. S15) and so do average annual temperature anomalies[27]. What are companies doing to lower their emissions? What differentiates companies that are successful in meeting climate targets from those that fail? Answering these questions requires a detailed knowledge of corporate sustainability behaviour and investment plans. Unfortunately, contrary to the disclosure of financial information, which is a strictly enforced and regulated process, the disclosure of nonfinancial information is largely voluntary and unregulated.

In recent years there has been substantial progress in standardising climate-related disclosure by, for example, the Task Force on Climate-Related Disclosure, TCFD. However, the proportion of companies following these standards is still limited (e.g., in 2021 only 4% of companies disclosed in line with all the recommended disclosures of the TCFD[28]). Therefore, information on corporate sustainability behaviour is scarce and difficult to quantify. To address this issue, in

this work we have presented a sustainability dataset that maps unstructured information contained in sustainability reports into a quantitative and systematic framework that can be used to study corporate sustainability behaviour. We have documented the evolution of corporate actions to limit emissions in the past ten years, and we have identified the behavioural differences that characterise companies with emissions pathways aligned with the targets set by the Paris Agreement.

Our analysis shows a large degree of heterogeneity in sustainability behaviour across companies in our sample, which includes some of the largest publicly traded corporations. Most sustainability initiatives focus on SDG 7 and 12 and involve risk mitigation strategies such as implementing changes in existing assets and procedures (Fig. 1). The low incidences of SDG 13 ("Climate Action") in our dataset, which could be surprising at first, is due to (1) the policy nature of SDG 13 targets and (2) the overlap in scope with other SDGs, most notably SDG 7 and 12. Focusing on companies in the hard-to-abate and energy-intensive sectors, we have found that the distribution of the number of initiatives across the sample is substantially skewed, with a limited number of companies taking a large number of initiatives compared to the sample mean (Fig. 2 panel b, S11). Interestingly, the relative incidence of the type of initiatives has changed substantially during our sample period. Specifically, we have found a rise and decline of initiatives aimed at implementing internal changes (such as modification of assets and procedures), and an opposite trend, with a recent increase, of innovation initiatives (Fig. 2 panel c). Yet, the former are still substantially more prevalent than the latter.

Looking at the link between sustainability behaviour and alignment with the climate targets set by the Paris Agreement, we have shown that, after accounting for asset characteristics, fixed effects, characteristics of the disclosure and self-selectivity, the total number of initiatives undertaken to lower emissions is unrelated to the magnitude of deviation and the probability of deviation of companies emission pathways from those compatible with the climate targets set by the Paris Agreement to lower global warming below 2 °C with respect to pre-industrial levels (Fig. 3 panel d). However, while the total number of initiatives cannot explain the alignment of emission pathways, the differential managerial effort placed on different activities/SDGs (i.e., the particular sustainability behaviour adopted by a company) is an important explanatory variable.

To illustrate this point, we have analysed the behavioural differences between aligned and misaligned companies. We have found that companies aligned with the Paris targets prioritise actions associated with growth opportunities, innovation (e.g., R&D investments, incentives, new products) and cooperation (association, communication). On the other hand, misaligned companies place relatively more effort in behaviours that prioritise risk management actions (e.g., asset modification, modification of procedures, assessment and measurements). Importantly, the differential behaviour is mostly driven by the goal of the actions, i.e., aligned companies focus substantially more on energy goals (SDG 7). Finally, we have shown that these behavioural differences can explain the observed differences in projected long-term emissions and the relative presence of renewable sources in the energy mix of power companies. Our main findings are shown in Figs. 4 and 5, Supplementary Figs. S14, S9–ST12 and ST3. Overall, our findings suggest that the misalignment between companies' climate goals, actions and outcomes is due to a widespread over-investment in risk mitigation actions as opposed to innovation and cooperation activities to foster energy goals.

As with any empirical work, there are several limitations of our analysis that can open up opportunities for future research. First, in our empirical approach, we have taken several precautionary steps to limit endogeneity issues and the impact of unobserved factors (see Material and Methods, section Sustainability initiatives and alignment with climate targets). However, because all our findings come from the

analysis of empirical data, we cannot assign a definite causal interpretation to our results. Specifically, our evidence points towards a statistically significant association between sustainability activities and goals (i.e., sustainability behaviour) and alignment with climate targets.

There are two possible interpretations for these associations. In the first interpretation, the associations imply a direct (causal) link between initiatives and alignment. Indeed, most of the initiatives reported in the analysis can be directly related to emission reduction processes. For example, we expect that developing new products and investing in research and development for developing and employing renewable energy sources will result in lower future emissions. Interestingly, however, as shown in Supplementary Table ST14, non-causative initiatives, such as donation & funding, also seem to play a role, albeit small. Therefore, our result can also imply the existence of a more complex latent causal structure. For example, companies that donate substantial capital to emission reduction activities might be more committed to sustainability issues and adopt behaviours that result in lower long-term emissions. In the second interpretation, our dataset provides a window to these latent causal structures (behavioural choices or sustainability strategies). Further research is needed to investigate this issue in further depth by, for example, explicitly identifying the latent causal structures.

Second, we do not differentiate between initiatives based on their complexity and funding structure because this differentiation would introduce an additional level of subjectivity in the codification of the training set. Importantly, we do not identify and distinguish greenwashing initiatives from effective actions. In our analysis, to limit the impact of green-washing on our results, we control for whether companies follow GRI standards in their reporting and if the reports have gone through an audit process. Following international standards and having reports verified by third parties limit the opportunities for green-washing, but further research is needed to identify greenwashing cases systematically and categorise initiatives based on the effort required to accomplish them.

Third, our sample includes companies from several countries with diverse regulatory frameworks and stakeholders' pressure on sustainability reporting, potentially influencing the characteristics of the sustainability reports and the disclosed behaviour. In our work, we account for differences in regulatory frameworks by controlling for country-level fixed effects. However, future studies can investigate in further depth the differences between the characteristics of the initiatives and the behaviour of companies subject to mandatory disclosure (such as companies in the European Union after the NFRD, Directive 2014/95/EU) versus those with operations in countries without existing regulatory frameworks.

Finally, even in countries with well-established non-financial disclosure regulatory frameworks, companies can hide their non-sustainability behaviour, i.e., activities that pose a high environmental risk. Characterising non-sustainability behaviour is beyond the scope of this manuscript. It is a complex process that requires the development of a theoretical framework and the collection of vast amounts of text from, for example, news media and legal databases. However, we believe it is a promising future avenue of research that can complement our approach.

## Implications of our work

There are several practical implications of our analysis and dataset, which we believe can be relevant for three societal actors. Firstly, business leaders can benefit from a detailed understanding of the sustainability behaviours of peers and competitors to improve their climate strategies. Our analysis already illustrates some results relevant for sustainability strategists, namely, the importance of focusing more on activities that create external value over those that involve changes in assets already in place.

Second, investors can use our datasets for allocating capital to its most sustainable use. Sustainable capital allocation requires market participants to have access to transparent information on the non-financial activities of public corporations[11,29]. However, this information is rarely available. Currently, investors mostly rely on Environmental ratings (the E-dimension and its sub-scores of ESG ratings) to assess the environmental sustainability of publicly traded companies. Our measure differs from environmental ratings both practically and conceptually. From a practical perspective, environmental ratings are subjective assessments of companies' exposure to environmental risks and are not necessarily predictive of future emissions reduction[30]. For example, Environmental ratings and their sub-scores cannot distinguish between companies with emission pathways aligned and misaligned with the climate target of the Paris Agreement (Supplementary Fig. S16 panel a). Also, they are uncorrelated with total emissions (Supplementary Fig. S16 panel b). On the other hand, our behavioural dataset focuses on the actions that companies are now taking to lower their environmental impact. Therefore it can be used to build predictive models grounded on transparent and objective information. From a conceptual perspective, environmental ratings are mostly derived from and used to measure outcomes (i.e., exposure to environmental risks). On the other hand, our approach was derived to measure effort in achieving sustainability targets (climate change SDGs, in the specific case of this work).

Finally, our dataset offers policymakers the opportunity to assess the status of sustainability reporting at large and to develop new regulations to improve transparency and reliability of nonfinancial reporting. Policymakers can also use our dataset to identify effective behaviours to incentives through policy and regulatory changes. Given the relevance of corporate behavioural changes to meet country-level targets[31,32], our dataset can be valuable to help nations to meet their nationally determined contributions.

Overall, this manuscript opens new opportunities for studying sustainability behaviour within a systematic and quantitative framework. It is the first of a series of studies that will use our corporate sustainability dataset as a key to understanding how to transform current business practices and align them to societal expectations. We believe this is a crucial step to foster the transition towards a more inclusive and just economy.

## Methods

### Behavioural dataset

Here we provide a brief summary of the process we follow to collect the behavioural dataset. In the Supplementary Section A we provide a more technical presentation and in Supplementary Fig. S1 we provide a schematic representation of the workflow. Our main unit of analysis is a sustainability initiative: a concrete action or set of related actions that a company is pursuing outside of its normal core business operations with the intent to directly address one of the 17 sustainable development goals (SDGs). Importantly, to be classified as an initiative an action need to refer to an activity that a company has done, or is actively pursuing. Investment plans and future projects are not regarded as initiatives.

Our study is centred around the analysis of annual corporate sustainability reports. Depending on availability, the sustainability reports are either standalone reports (i.e., reports that only present non-financial information), integrated reports (i.e., reports that present financial and non-financial information within an integrated framework), or annual reports with a significant section on sustainability. The links to the PDFs are from REFINITIV. The sustainability reports that were not available at the URLs provided by REFINITIV were bought from Corporate Register (https://www.corporateregister.com/). All reports that were not available from either REFINITIV nor Corporate Register were crawled from the internet. Overall, we analyse 32183 reports for 7235 companies from 2000 to 2020. Out of these

companies, we were able to match the behavioural dataset with complete information about emissions and accounting data in the observation period 2010-2020 for 4191 companies and 18719 reports. Of these 4191 companies, 1951 are part of the hard-to-abate and energy-intensive sectors (Energy, Utilities, Material and Industrial in the GICS classification) analysed in the main text. Supplementary Fig. S4, Supplementary Tables ST4 and ST5 show the summary statistics of our population while Supplementary Fig. S5 shows the summary statistics of the full population before matching the behavioural dataset with accounting and emission data.

In order to extract sustainability initiatives from the texts of the reports we use neural machine learning models trained on a training set developed by the GOLDEN Foundation (http://foundationgolden.org/blog/golden-is-golden-for-impact/), see Supplementary Section A in the Supplementary Information. The training set was created by manually annotating 507 sustainability reports (~55088 initiatives). In the annotation process, the annotators were instructed and trained by two of us to identify initiatives as implemented actions and to consider commitments and plans as not-an-initiative. The annotators were also instructed and trained on mapping sustainability objectives to the most closely related SDGs (following the official definitions that can be found at https://sdgs.un.org/goals) and activities to our classification scheme. In the Supplementary Section B, we provide a detailed definition of the activities. The classification scheme is generated by reading sustainability reports and identifying common activities described by the corporations. While there could be alternative taxonomies to classify the activities, we believe that those reported in the main text are the most common mutually exclusive and collectively exhaustive activities pursued by the companies in our sample.

To extract the initiative from the text in the data-generating process, the documents are converted from pdf formats to json, making them machine-readable. Textual fields from the pdfs are extracted and converted to plain text. The full text is separated into individual sentences for further analysis. Metadata from the pdf is also extracted, such as the creation time and any optional comments that were added by the authors. The system analyses each sentence in every report in order to determine whether they refer to sustainability initiatives. Sentences classified as an initiative are then further combined with their preceding and subsequent two sentences (i.e., their context), as a single initiative is often described with multiple sentences or whole paragraphs. This process helps us reduce the possibility that two adjacent sentences, belonging to the same initiative, are double-counted in our dataset. We use two separate machine learning systems for this task (a BERT and a RoBERTa-based model) and combine their predictions together for an ensemble model in order to achieve the best accuracy (see Supplementary Section A). We use a BERT and a RoBERTa-based model because of their capacity to interpret words and sentences within their context[33,34], which is an essential requirement for our task. After the algorithm identifies an initiative, the text goes through a separate system that classifies it within its context based on (1) the type of the action or activity (e.g., adoption of standards and rules, communication, donation & funding, etc.) and (2) the most closely related Sustainable Development Goal (SDG). To assess the ex-post quality of the dataset we perform a manual check of a random sample of initiatives to assess the precision of the classification in the correct activity type (87%) and SDG (86%), see Supplementary Section A.4 in the Supplementary Information.

While our dataset covers the full spectrum of sustainability initiatives, in the main text we focus solely on those initiatives that address the problem of reducing GHG emissions. To isolate these initiatives from the rest we analyse the text extracted from the reports and we only keep those initiatives that mention: climate change, emissions, global warming, greenhouse gases (or ghg), green technologies, renewable, energy efficiency, environmentally efficient, natural energy, fuel-efficient, electric power consumption, energy use,

energy saving, carbon reduction, energy consumption. To identify the words in the dictionary we first start with a few keywords (climate change, emissions, global warming). Then we isolate initiatives containing those words and we look extensively at all the other initiatives. From these other initiatives we select a second subsample and repeat the process until the discarded initiatives do not contain a significant number of actions aimed at reducing emissions (see Supplementary Section C for example of initiatives).

In part of our analysis we refer to different types of actions as being part of a broader macro categorisation (e.g., innovation). For clarity, here we report the macro categorisation which follows a similar logic as that of the empirical mechanisms outlined in ref. 24. Specifically, Innovation actions are: 'association', 'r&d investments', 'new products'. Internal changes actions are risk-mitigating activities such as: 'adoption of standards and rules', 'organizational structuring', 'assessment and measurement', 'modification of procedures', 'asset modification', and 'training'. Finally, Reputation and stakeholder engagement actions are: 'communication', 'pricing', 'incentives', 'volunteerism', 'donation & funding'.

## Fundamental and emission data

Additionally to our behavioural dataset, in this work we use data from third parties. Specifically, we use COMPUSTAT for company fundamentals. We define Size as the log of sales (SALE, in USD) adjusted for inflation (https://fred.stlouisfed.org/series/CPIAUCSL); Invested Capital is long plus short-term debt (DLTT+ DLC), plus book equity (CEQ) plus cash and short-term investments (CHE); Tangibility is property plant and equipment (PPENT, in USD) divided by book assets (AT, in USD). Exchange rates are from REFINITIV. Information on the characteristics of the reports (if they follow GRI guidelines and are assured by external audit firms) and the links to the pdfs are from REFINITIV Asset4. Equity data used to calculate total market capitalisation are from REFINITIV. Finally, data for global GHG emissions are from the climate watch portal (https://www.climatewatchdata.org/).

Company-level emissions data are from Trucost. In particular we measure total GHG emissions as Direct plus first-tier indirect emissions which are defined as GHG protocol scope 1 emissions, plus any other emissions derived from a wider range of GHGs relevant to a company's operations, plus GHG protocol scope 2 emissions, plus the company's first-tier upstream supply chain. This is the Trucost's default measure of emissions (see https://www.spglobal.com/spdji/en/documents/additional-material/faq-TruCost.pdf). Emission data from Trucost are a combination of self-reported and estimated data. In our sample, approximately 65% of the Scope 1 and Scope 2 emissions values are self-reported. On the other hand, approximately 55% of the the Scope 3 emissions data are self-reported.

## Climate targets

Data on alignment with climate targets are from Trucost. Specifically, we use the difference between the projected emission pathway of a company as of 2018, 2019, and 2020 and the required pathway to limit global warming below 2 °C. Additionally, Trucost also provides data on alignment with a "well below" 2 °C outcome. In the Supplementary Information we use these data to confirm the validity of our results. Negative deviations indicate alignment with the climate targets, and positive deviations indicate misalignment. Trucost estimate the emission pathway using the methodologies highlighted by the Science Based Targets Initiative (SBTI). Specifically, they use the Sectoral Decarbonization Approach (SDA) for high-emitting companies with homogeneous business activity and the Greenhouse Gas Emissions per Unit of Value Added (GEVA) approach for low-emitting companies with heterogeneous business activities. They also use two additional models, namely the "GEVA Modelled Approach" and the "GEVA Modelled including Constant Intensity Approach" for companies that do not disclose all the relevant emission information and that therefore

require modelling. Importantly, companies that do not disclose the information required by the Trucost input-output models are excluded from the universe.

The analyses discussed the main text focus on the study of the differential behaviour of companies in the aligned and misaligned population. Differential behaviour is measured as excess effort along the different behavioural dimensions. Excess effort of the aligned population is measured as the average difference between the normalised behavioural matrices $\mathcal{B}$ in the aligned and misaligned population. We normalise the behavioural matrix of company $i$ in year $t$ by dividing each element of the matrix by the total number of initiatives of company $i$ in year $t$.

### Sustainability initiatives and alignment with climate targets

To estimate the association between the total number of initiatives and companies' alignment with climate targets, we estimate two regression models. In the first specification, we use the magnitude of deviation from the target as the dependent variable in a linear model that includes as control (independent) variables the total number of initiatives (or the initiatives in positive or negative excess effort) and all those factors that could confound the estimations (i.e., factors that are associated with both the number of initiatives and the magnitude of deviation). In the second specification, we use the same control variables, but we estimate a Probit model using an indicator variable for the alignment with climate targets as the dependent variable.

In both specifications, the control set includes Revenue, Invested Capital, the proportion of tangible assets (Tangibility) and GHG emissions. Companies can use revenue and invested capital to finance sustainability initiatives but also activities that generate GHG emissions. Therefore they are the crucial confounder of the association between initiatives and magnitude/probability of alignment. Tangibility also acts as a confounder because property plants and equipment generate emissions but also require initiatives for maintenance and upgrade (e.g., asset modification). GHG emissions are a crucial input in the estimation of alignment, and they are also strongly positively correlated with the total number of initiatives (through a size effect). Therefore, they also are an essential confounder to control for. We measure the control variables on a rolling basis (we take historical averages) because it is not the value of initiatives or the assets' characteristics at time $T$ that influence the magnitude/probability of alignment, but rather their historical values characterise the capacity of companies to have aligned emission pathways. Taking historical averages of the control variables also helps us address the problem of reverse causality, i.e., we want to show that it is the average behaviour in the years before estimating the emission pathways that facilitate the alignment.

In addition to the asset characteristics, we also control for time-invariant factors, which include country and sector fixed effects. We do not control for firm and year-fixed effects because the regression is estimated with rolling average factors in the year of estimation of the alignment. Country-fixed effects are necessary to account for systematic differences in regulatory frameworks to which companies in our sample are subject. This is an important control factor because the regulatory frameworks on climate policies and sustainability reporting can change significantly from country to country, and these differences can influence the information content of the reports. Sector fixed effects instead are necessary to account for differences in technological basis across companies.

Next, given the discretionary nature of sustainability reporting, we control for two important characteristics of the reports that are potentially related to the quality of the disclosure. Specifically, we include fixed effects to (1) distinguish reports that follow GRI guidelines versus those that do not and (2) distinguish reports that are subject to an external audit process. Because the behaviour is

estimated on a rolling basis, we add the fixed effects for each year in the window. For each report characteristic, we observe three classes: the report displays the characteristics, the report does not display the characteristic, and the data is missing. Missing data occur when Asset4 analysts are not able to conclusively assert that the report display or does not display specific characteristics. This ambiguity could impact the result of our analysis. Therefore, we run a test to assess the robustness of the result due to this possible source of error in the control variables. Specifically, we repeat the model estimation by assuming that missing information corresponds to a lack of the specific characteristic, and instead of fixed effects, we include in the regressions an indicator variable that takes the value of one if the report displays a particular characteristic and zero otherwise. Results are shown in the Supplementary Table ST8 and are qualitatively unchanged (i.e., the magnitude of the coefficients is slightly different but their sign and significance is the same).

As discussed in section Climate targets, companies' alignment with climate targets is calculated using either self-reported emissions or emissions estimated by Trucost, depending on data availability. Because Trucost estimates are noisier than self-reported emissions measures, there could be a bias induced by the nature of the noise. To control for this possible source of bias, we add an indicator variable that takes the value of one if Trucost uses self-reported data to estimate alignment and zero otherwise.

Because companies that issue sustainability reports might differ systematically from non-publishing companies, but we only observed the reports for the companies that issue one, there is the potential for self-selectivity in our sample. To address this source of endogeneity, we estimate the model using the Heckman correction[26]. That is, first, we run a Probit model where the independent variable is one if a company issues a report in year $Y$ and zero otherwise. Data on the issuance of sustainability reports are from REFINITIV. The independent variables include Size, Tangibility and Invested Capital, as well as the proportion of companies in the same sector and country that also issue a sustainability report in year $Y$. The coefficients of the Probit model are shown in Supplementary Table ST6 in the Supplementary Information. From the fitted Probit we estimate the inverse Mills ratio, $\mathcal{M}$, which is defined as: $\mathcal{M} = \frac{f(x)}{1-F(x)}$ where $f(x)$, $F(x)$ are the (normal) probability density function and the cumulative distribution, respectively. Then we use the inverse Mills ratio from the Probit as an additional covariate in the main specifications. Standard errors are adjusted for heteroskedasticity in all the regressions.

In summary, we estimate the following two models:

$$\text{Magnitude of deviation: } \mathcal{D}_{i,T} = \alpha + \beta \mathcal{I}_{i,\langle t\rangle_2} + \sum_j \gamma_j \mathcal{X}_{j,i,\langle t\rangle_3} + \boldsymbol{\delta}\mathbf{C}_i$$
$$+ \boldsymbol{\eta}\mathbf{S}_i + \xi\mathcal{M}_i + \boldsymbol{\mu}\mathbf{R}_i + \phi\mathbb{1}_{i,e} + \epsilon_i$$

$$\text{Probability of deviation: } \mathbb{P}(\mathcal{A}_{i,T}=1) = \alpha + \beta \mathcal{I}_{i,\langle t\rangle_2} + \sum_j \gamma_j \mathcal{X}_{j,i,\langle t\rangle_3} + \boldsymbol{\delta}\mathbf{C}_i$$
$$+ \boldsymbol{\eta}\mathbf{S}_i + \boldsymbol{\mu}\mathbf{R}_i + \phi\mathbb{1}_{i,e} + \epsilon_i$$

$$(1)$$

Where $\mathcal{D}_{i,T}$ and $\mathbb{P}(\mathcal{A}_{i,T}=1)$ are the magnitude of deviation from the target (negative values indicate alignment) and a binary indicator for alignment for company $i$ as estimated at time $T$, respectively. $\mathcal{I}_{i,\langle t\rangle_2}$ and $\mathcal{X}_{j,i,\langle t\rangle_3}$ are the number of initiatives and the control factors of firm $i$ estimated on a rolling window. $\mathbf{C}$ and $\mathbf{S}$ are country and sector fixed effects, $\mathcal{M}$ is the inverse Mills Ratio, $\mathbf{R}$ are the report characteristics fixed effects in the estimation window of the behaviour, and $\mathbb{1}_e$ is the indicator variable for the source of emission data.

### Topic analysis

Figure 6 aggregates results obtained from a topic analysis we performed on the text of the sustainability initiatives. Starting from the

joint text of all initiatives performed by each company in a given year, we build a document-term matrix whose rows are company-year observations and each column is associated with a word frequently encountered across companies (all non-stopwords that appear more than 200 times). Each cell of the matrix corresponds the number of times the column-word is present in the joint text associated with the row-observation. We then probe the underlying structure by imputing the correlation between the occurrence patterns of different words to the fact that they belong to the same topic[35,36]. We employ an Hamiltionian Monte Carlo estimation method to uncover the combination of words into topics that best fit the data[37], and then interpret the topics manually starting from their most defining words[38]. Supplementary Table ST13 shows the words in each topic in descending order of appearance probability and the title we associated with them.

We then focus on the posterior probability distribution of topics across documents. Specifically, our analysis assigns to a company-year joint texts a set of probability values for each topic. These values signify the relative prevalence of words associated with different topics in the joint text, weighed by the probability of the words actually belonging to those topics. This allows us to identify which topics companies were more likely to talk about with respect to each other. In Fig. 6, we average this topic prevalence variable among aligned and misaligned companies, uncovering the significant language difference between the two groups. In Supplementary Figs. ST17 and ST18, we performed additional tests for the robustness of the results of the topic analysis.

### Reporting summary

Further information on research design is available in the Nature Portfolio Reporting Summary linked to this article.

## Data availability

The behavioural dataset generated in this study has been deposited in Harvard Dataverse at https://doi.org/10.7910/DVN/C7ILED[39]. Data from COMPUSTAT, Trucost, and Refinitiv can be accessed directly from the data providers for a fee, see refs. 40–44

## Code availability

The code to reproduce our results are publicly available on Harvard Dataverse at https://doi.org/10.7910/DVN/C7ILED[39].

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

## Acknowledgements
The authors would like to thank Marina Gadkary, Vincenzo Vastola, Emanuele Bettinazzi, Peter Snoren, Joeri Rogelj, Livio Scalvini and the members of the "GOLDEN For Impact" research committee for their help in the development of the dataset and for fruitful discussions.

## Author contributions
S.C. and M.B. designed the study, S.C. performed the analysis and wrote the first draft of the paper. M.R. developed the algorithm to generate the data. M.Z. developed the theoretical framework for the generation of the dataset. M.B. supported the development of the dataset. S.C., M.B., M.R., M.Z. wrote the final version of the paper.

## Competing interests
The authors declare no competing interests.
