## [Peer Review File · Nature Communications]

The alignment of companies' sustainability behavior and emissions with global climate targetsREVIEWER COMMENTS

Reviewer #1 (Remarks to the Author):

A) Summary:

The manuscript under review presents a systematic framework for transferring unstructured textual data into information beneficial for environmental risk analysis. The inventive aspect is reflected in the regression model designed as a fusion of components derived from classifying actions and activities pursued by corporations.

B) Reasons to Accept:

The manuscript modernizes the framework for handling textual data by integrating advanced deep learning techniques, particularly those based on BERT-based pretrained-models. It provides a noteworthy contribution through the introduction and analysis of a unique dataset.

C) Reasons to Reject:

1) Concerns Regarding Clarity in Writing:

- 1.1. The technical workflow, as currently explained, is challenging to comprehend. It would be beneficial if 2-3 sentences were included in the abstract and 3-5 in the introduction, outlining the workflow.
- 1.2. In "Introduction" (Page 3), given the complexity of data preprocessing and the system's multiple modules, a detailed figure should be provided illustrating the workflow and where each component fits within the overall manuscript.
- 1.3. "A Model for Sustainability Initiatives" (Page 23) should better articulate the origin of each component in the regression model formula, rather than placing these descriptions afterwards.

2) Technical Discrepancies:

- 2.1. The topic analysis would benefit from the inclusion of various techniques such as K-means or t-SNE. A more rigorous quantitative analysis is warranted.
- 2.2. In "Extraction of Initiative Sentences" (Supplementary Information Section A), the rationale for the 0.66 threshold as optimal is not clear.
- 2.3. For querying initiative sentences associated with the Paris Rules, the application of retrieval models like BERT Dense Retrieval [1] should be considered.
- 2.4. A comparison of the proposed methods with other benchmarks, such as [2], would enhance the manuscript's credibility.

3) Motivational Aspects:

- 3.1. The manuscript lacks in-depth analysis of corporations' related-hazard actions and their environmental impact. It is also missing a critical evaluation of which categories pose the greatest risk to the environment and why.
- 3.2. The study would be more insightful if it could identify and critically evaluate which categories of corporate activities pose the highest risk to the environment and provide justifications for these conclusions.

References:

- [1] Shengyao Zhuang, Guido Zuccon. TILDE: Term Independent Likelihood model for Passage Re-ranking. SIGIR. 2021.
- [2] Julian F Kölbl and others, Ask BERT: How Regulatory Disclosure of Transition and Physical Climate Risks Affects the CDS Term Structure, Journal of Financial Econometrics. 2022.
- [3] July Bias Macêdo, Márcio das Chagas Moura, Diego Aichele, Isis Didier Lins, Identification of risk features using text mining and BERT-based models: Application to an oil refinery. Process Safety and

Reviewer #2 (Remarks to the Author):

The authors use a machine learning algorithm to analyse the contents of corporate sustainability reports in order to extract, classify and quantify unstructured, textual information about corporate sustainability initiatives and their alignment with SDG goals. The information about corporate initiatives (climate actions) is then mapped onto the TruCost data about aligning the GHG emissions reduction pathways with the goals of the Paris agreement. The authors proceed with analysing behavioural differences between companies that are aligned and "misaligned" with the climate targets and find that the former group of companies tend to pursue innovation-, growth- and cooperation-related initiatives, the latter focuses on more internal-oriented, risk mitigation procedures. The study is very thorough, and the authors have done an impressive work in building a unique dataset based on nonfinancial corporate information. The methodology is sound and the choice of measures is well-explained and well-motivated. I particularly appreciated the different "levels" of analysis on different subsets of data which allowed making meaningful comparisons and draw interesting conclusions.

Yet, my impression is that a few points need to be clarified before recommending accepting the paper for publication.

First, while supplementary materials clearly describe how sustainability activities (e.g., R&D investments, donations and funding, asset modification) are identified from the sustainability reports and how they are linked to achieving the climate targets of the Paris agreement, it is less clear how SDGs matter for the goals of the paper. As the authors mention on page 7, few SDG targets are directly related to GHG emissions, and the most relevant goal SDG 13 is difficult to tie to corporate-level initiatives. In fact, with the exception of SDG 7 and 12, other SDGs' importance seems to be rather marginal in the paper. Furthermore, companies have been criticized for engaging very superficially with SDGs by "cherry-picking" only those SDGs that are easy for them to achieve, lacking substance in reporting on SDGs and providing vague statements about the relation of their activities to the SDGs. For the reasons mentioned above, I would suggest making a stronger case for including SDGs alignment in the paper.

Second - and somewhat related to the point raised above - the authors mention that part of the novelty of their study is that they look at the "implemented actions" (p. 3) as opposed to the stated commitments of companies. However, information contained in sustainability reports may be used for impression management purposes and may not adequately reflect the actual sustainability-related internal management practices, decisions and actions. So, I was wondering how does the algorithm deal with "green washing" cases? I am not suggesting that the algorithm is supposed to recognize such cases but it warrants a discussion, in my view. On page 11, the authors mention that they "control for self-selectivity on the disclosure of the emissions" without specifying how this happens (unless I have overlooked something). Finally, is algorithm able to distinguish between different time orientation (past- vs. future-oriented) in the reports ("we have commenced an energy consumption monitoring program" vs. "we have decided to commence an energy consumption monitoring program")?

Third, the authors include companies from different countries in the dataset which operate within different national legal frameworks when it comes to nonfinancial reporting. For instance, during the time period considered by the study, the nonfinancial reporting directive (NFRD, Directive 2014/95/EU) in the European Union has made it mandatory for the large companies with over 500

employees to report on environmental, social and governance matters. Hence, the criteria for nonfinancial reporting are different than for other companies where such regulation does not apply. The authors might consider an additional type of analysis to understand if the results are qualitatively similar in mandatory vs. voluntary sustainability reporting settings.

Reviewer #3 (Remarks to the Author):

Summary

This paper develops a novel proxy for corporate initiatives based on automated textual analysis of corporate sustainability reports. The authors focus on correlations between their novel proxy and corporate emissions, with a focus on alignment with Paris Agreement emissions trajectories. Emissions data comes from TruCost, a third-party data vendor.

The main results are twofold: first, the total number of initiatives is not significantly associated with the level of emissions or Paris alignment. Second, and perhaps more interesting, certain types of initiatives are positively associated with Paris Alignment, while others are negatively associated.

The authors frame their contribution in terms of developing a measure and framework that can help guide capital allocation and policy.

Evaluation

The paper presents interesting novel data, and a potentially useful way of analyzing texts from sustainability disclosures. However, I think the contribution of the paper may be limited relative to the bar for Nature Communications. My concerns relate primarily to measurement, causality, and scope.

1. Measurement: Unfortunately, there are important unaddressed issues in both the primary independent and dependent variables.

a. Sustainability reports: These are voluntary disclosures, such that firms have significant latitude in what they contain. Often, these are glossy, almost marketing documents. Firms also have significant discretion in how they portray any initiative. Furthermore, in measuring initiatives, I was surprised that the authors chose to only look at three consecutive sentences. Often, firms will devote entire pages to initiatives, including pictures and even plots. Overall, I am concerned that the authors have not sufficiently considered how the discretionary nature of the content of sustainability reports might affect the properties of measured initiatives, and how that might filter into associations with emissions and Paris Alignment. I was also surprised by statistics on the sustainability reports. They seem to diverge quite a bit from those provided by Gipper et al. (https://papers.ssrn.com/sol3/papers.cfm?abstract_id=4263085).

b. Emissions: TruCost sources emissions information from firm disclosures and proprietary estimates. If coming from firm disclosures, the authors should consider how colocation in the same document as the initiatives might affect associations. If coming from estimates, the authors need to consider what is driving the firm to provide insufficient emissions disclosures such that TruCost needs to estimate emissions. Additionally, TruCost estimates are likely to be noisier than firm-derived emissions measures, which could systematically bias the analysis of associations between initiatives and emissions. Specifically, firms that provide less information about initiatives are also likely to provide less information about emissions, though there may be other selection mechanisms at play.

2. Causality and scope: The authors are clear that readers should not infer a causal relation from their analysis. However, the causal relation is presumably the most important part. I was left curious about whether the initiatives facilitated alignment or if firms with aligned objectives chose to engage in systematically different types of initiatives. I am not sure that showing a correlation between types of initiatives and Paris Alignment provides a sufficient contribution. A high-level contribution might come from showing what types of initiatives drive emissions decreases that facilitate alignment, but that does not seem to be feasible from the current setting.

Response to Reviewers: Sustainability behaviour of high-emitting companies and their alignment with global climate targets

Simone Cenci¹, Matteo Burato¹, Marek Rei^{1,2} and Maurizio Zollo¹

¹Leonardo Centre on Business for Society, Imperial College Business School, London, UK

²Department of Computing, Imperial College London, London, UK

Reviewer 1

- A) Summary:

The manuscript under review presents a systematic framework for transferring unstructured textual data into information beneficial for environmental risk analysis. The inventive aspect is reflected in the regression model designed as a fusion of components derived from classifying actions and activities pursued by corporations. The manuscript modernizes the framework for handling textual data by integrating advanced deep learning techniques, particularly those based on BERT-based pretrained-models. It provides a noteworthy contribution through the introduction and analysis of a unique dataset.

Answer. Thank you for taking the time to review our manuscript. We appreciated your comments and made several changes to the manuscript to address your concerns. Below we respond to your comments, and we point to new discussions and analyses in the new version of the manuscript that we hope address your concerns.

- 1) Concerns Regarding Clarity in Writing: 1.1. *The technical workflow, as currently explained, is challenging to comprehend. It would be beneficial if 2-3 sentences were included in the abstract and 3-5 in the introduction, outlining the workflow.*

Answer. Thank you for the suggestions. We fully agree that more clarity in the introduction can benefit the broad readership of Nature Communications. Therefore, to help the reader better understand our estimation process without the need to read the full description in the Supplementary Information, we have followed your recommendation and added more details on the workflow in the introduction (see lines 70-86 in the new version of the manuscript). Please note that we provide a more in-depth description in the Material and Methods section "Behavioural dataset" and the Supplementary Information section A. Given that Nature Communications is a general audience journal, we have preferred to avoid including technical descriptions in the abstract.

- 1.2. *In "Introduction" (Page 3), given the complexity of data preprocessing and the system's multiple modules, a detailed figure should be provided illustrating the workflow and where each component fits within the overall manuscript.*

Answer. Thank you again for the great suggestion. We have now made a figure to illustrate our data-generating process (Figure S1 in the new version of the manuscript). We agree that such a figure can help the reader understanding our approach.

- 1.3. *"A Model for Sustainability Initiatives" (Page 23) should better articulate the origin of each component in the regression model formula, rather than placing these descriptions afterwards.*

Answer. Sorry for the lack of clarity and thank you for the useful recommendation. We have now greatly expanded the section and the description of the model by (a) articulating the reasoning we followed to develop it; (b) describing the origin of each component in detail, and (c) presenting the equations and explaining each symbol individually. You can find the new section "Sustainability initiatives and alignment with climate targets" in page 26, line 660 of the new version of the manuscript. We believe the new description is significantly clearer.

- 2) *Technical Discrepancies: 2.1. The topic analysis would benefit from the inclusion of various techniques such as K-means or t-SNE. A more rigorous quantitative analysis is warranted*

Answer. Thank you for the suggestion. We complemented our topic discussion with the use of (1) a t-SNE method and related visualization, and (2) silhouette analysis of implied initiative clustering by topic. We report both analyses in the Supplementary Information (Figure S17) and reference them in the main text (lines 759-760). We also employ the t-SNE framework in liaison with Figure 5 by investigating the distinctiveness of relatively misaligned vs relatively aligned topics in Figure S18. These additional robustness checks generally reinforce the reliability of the identified topics and confirm the distinctiveness of the misaligned and aligned topics of Figure 5.

- 2.2. *In "Extraction of Initiative Sentences" (Supplementary Information Section A), the rationale for the 0.66 threshold as optimal is not clear.*

Answer. Thanks for pointing out that this point requires more explanations. The threshold was manually tuned based on the automatic evaluation of the development set and the manual evaluation of an output sample check. The goal of the exercise was to find a trade-off between precision and recall of the identification of initiatives. We have set a threshold (0.66) that resulted in a 95% precision in the identification of the initiatives. We have tuned the algorithm to maximise precision because we wanted to ensure that the final dataset included only actions effectively implemented by firms in our sample (see also the importance of this choice in answer to Reviewer 2). We have now provided more details on this process in section "A.2 Initiative detection" in the Supplementary Information of the new version of the manuscript.

- 2.3. *For querying initiative sentences associated with the Paris Rules, the application of retrieval models like BERT Dense Retrieval [1] should be considered.*

Answer. Apologies for the confusion. TILDE is an information retrieval model designed to find documents that are similar to a given query. In our approach, we do not simply look for sentences associated with a specific query (the targets of the Paris Agreement). Instead, we developed a training dataset, and we trained a machine learning model to identify parts of the text containing any action that describes a sustainability initiative, which may be expressed in a wide variety of ways. Simply retrieving sentences that mention some form of sustainability would not be accurate since such a process would not allow us to distinguish initiatives from commitments or plans, a distinction that is at the core of our process. Therefore, we use a supervised classifier instead. We have clarified this point in section "Behavioural Dataset"

in the new version of the manuscript. Note as well that we do also make use of the BERT model, which is the main component of TILDE.

- 2.4. *A comparison of the proposed methods with other benchmarks, such as [2], would enhance the manuscript's credibility.*

Answer. The referenced paper trains the BERT model, which is also an important component in our system. However, as the tasks are quite different, the two resulting systems are not directly comparable. In addition to classifying text according to different dimensions, our system is designed to identify full initiative descriptions, which can consist of whole spans of text (I.e., multiple sentences). In contrast, Kölbl et al. focus only on classifying individual sentences. However, we are now citing this work and other works that make use of BERT to identify some information content in sustainability reports so that the reader can better contextualise our work within the current literature (see section "Behavioural Dataset", lines 587-589 in the new version of the manuscript). Thanks for the suggestion.

- 3) *Motivational Aspects: 3.1. The manuscript lacks in-depth analysis of corporations' related-hazard actions and their environmental impact. It is also missing a critical evaluation of which categories pose the greatest risk to the environment and why.*

3.2. *The study would be more insightful if it could identify and critically evaluate which categories of corporate activities pose the highest risk to the environment and provide justifications for these conclusions.*

Answer. We pooled these two points together because they touch on the same limitation, which is a great opportunity for further work. In this manuscript, we focus exclusively on corporate actions aimed at addressing global environmental sustainability challenges, particularly climate change, through analysing the information content of sustainability reports. The analysis required a significant investment for creating (1) a theoretical framework, (2) the manually annotated dataset to train the algorithms, and (3) a process to find/buy tens of thousands of reports, storing them and processing them. The unique aspect of the dataset is that it identifies and, importantly, systematically categorises corporate actions not only according to their goals (SDGs) but critically to their behavioural content.

As with any other work, there are limitations, one of which is, as you correctly pointed out, that we do not analyse the corporations' related hazard actions. Analysing the private sector's hazardous behaviours is a different, extremely important, question. There is no doubt that this is an essential line of work, but it differs from the one we have addressed in this manuscript. This analysis requires identifying alternative bodies of text (as environmental controversies are not disclosed in publicly available sustainability reports) and developing a theoretical framework and a large training set to extract their information content. Please note that the intersection between the two lines of work, including data from both the "bright" and the "dark" side of business behaviour, is undoubtedly part of our research agenda, and we have now addressed in the "Discussion" section this point as fruitful direction for future work (see lines 490-496 in the new version of the manuscript). This manuscript is a necessary step in that direction.

Reviewer 2

• *The authors use a machine learning algorithm to analyse the contents of corporate sustainability reports in order to extract, classify and quantify unstructured, textual information about corporate sustainability initiatives and their alignment with SDG goals. The information about corporate initiatives (climate actions) is then mapped onto the TruCost data about aligning the GHG emissions reduction pathways with the goals of the Paris agreement. The authors proceed with analysing behavioural differences between companies that are aligned and “misaligned” with the climate targets and find that the former group of companies tend to pursue innovation-, growth- and cooperation-related initiatives, the latter focuses on more internal-oriented, risk mitigation procedures. The study is very thorough, and the authors have done an impressive work in building a unique dataset based on nonfinancial corporate information. The methodology is sound and the choice of measures is well-explained and well-motivated. I particularly appreciated the different “levels” of analysis on different subsets of data which allowed making meaningful comparisons and draw interesting conclusions.*

Yet, my impression is that a few points need to be clarified before recommending accepting the paper for publication.

Answer. Thank you for taking the time to review our manuscript and for the thoughtful comments and suggestions. We believe your recommendations have helped us strengthen the message of the paper. Below we respond to your comments and point to new discussions and analyses that we have included in the new version of the manuscript to address your concerns.

• *First, while supplementary materials clearly describe how sustainability activities (e.g., R&D investments, donations and funding, asset modification) are identified from the sustainability reports and how they are linked to achieving the climate targets of the Paris agreement, it is less clear how SDGs matter for the goals of the paper. As the authors mention on page 7, few SDG targets are directly related to GHG emissions, and the most relevant goal SDG 13 is difficult to tie to corporate-level initiatives. In fact, with the exception of SDG 7 and 12, other SDGs’ importance seems to be rather marginal in the paper. Furthermore, companies have been criticized for engaging very superficially with SDGs by “cherry-picking” only those SDGs that are easy for them to achieve, lacking substance in reporting on SDGs and providing vague statements about the relation of their activities to the SDGs. For the reasons mentioned above, I would suggest making a stronger case for including SDGs alignment in the paper.*

Answer. Following the behavioural science literature, we characterise corporate behaviour as a combination of actions and goals. For the actions, we have developed a categorisation scheme generated by reading sustainability reports and identifying common activities described by the corporations. For the goals, we have decided to follow the language of the SDGs as goal setting framework to align our dataset with current trends (1) in corporate reporting and (2) in climate policies. Note that the classification along the SDG dimensions is performed based on the most closely related SDG associated with the stated objective, not on the company’s self-reported statement on SDG alignment. For example, an investment in research and development (the activity) aimed at increasing the energy efficiency of a particular production process would be classified as an activity that targets SDG 12, even if the company does not directly refer to SDG 12 in their reporting. The sustainability goal

- SDG mapping is codified within the training set in the same fashion as the sustainability activities. We have now clarified this point along the text and, in particular, in lines 73-76 and in lines 562-567 of the new version of the manuscript.

In terms of descriptive statistics, we believe it is interesting to simply show how companies distribute their effort along the different sustainability challenges (figure 1). We believe tracking this distribution can be extremely relevant to monitor companies' efforts to contribute to sustainable development. However, you are correct, companies have been criticised for engaging superficially with sustainability goals, and the information content of this classification can, in principle, be minimal. This is a fair objection, and therefore we have performed an additional analysis to support our choice of including SDGs alignment in the paper.

To assess whether the SDG classification adds value to our analysis, we performed an additional test. Specifically, we have re-estimated the main regression specifications (figure 4 panel c), excluding the SDGs. That is, we constructed the positive and negative excess effort variables (the relevant independent variables in the model) by only looking at the differential behaviour along the activity types and re-estimated the association of these variables with the magnitude and probability of alignment with climate targets. The rationale of the test is that if the SDGs dimensions contain relevant information, then the results of these new specifications should differ significantly from those reported in the main text in Figure 4 panel c, which includes both dimensions of behaviour. If the results do not change, then there would be limited reasons to include SDGs alignment in the paper.

The results are shown in Table ST8 in the new version of the manuscript, Supplementary Information. The table shows that SDG alignment contains relevant information, as the association between behaviour and alignment is substantially lower when we exclude the differential behaviour across the SDG dimensions from the analysis. To further investigate the importance of the two dimensions of behaviour (activities and goals), we have then repeated the same test by ignoring the differential behaviour along the activity types. Results are weaker but still marginally statistically significant. Overall, the result of this test illustrates the importance of looking at behaviour from the perspective of both the what (activity) and the why (goal) of corporate climate actions. You can find a discussion on this important test in lines 350-352 of the new version of the manuscript.

• *Second - and somewhat related to the point raised above – the authors mention that part of the novelty of their study is that they look at the “implemented actions” (p. 3) as opposed to the stated commitments of companies. However, information contained in sustainability reports may be used for impression management purposes and may not adequately reflect the actual sustainability-related internal management practices, decisions and actions. So, I was wondering how does the algorithm deal with “green washing” cases? I am not suggesting that the algorithm is supposed to recognize such cases but it warrants a discussion, in my view. Finally, is algorithm able to distinguish between different time orientation (past- vs. future-oriented) in the reports (“we have commenced an energy consumption monitoring program” vs. “we have decided to commence an energy consumption monitoring program”)?*

Answer. Thank you for pointing out that this point needs clarification. To deal with time orientation we codified the distinction between stated commitments (i.e.,

forward-looking statements of commitments and plans) and implemented actions (i.e., completed and/or in process activities) in the training set which is a standard approach in NLP (see line 563-565 of the new version of the manuscript). An initiative is defined as an implemented action, and annotators have followed this definition to label sentences. Once this distinction is codified in the training set, the algorithm is trained to learn how to differentiate the two. To ensure that the algorithm's output preserves this distinction, we have also followed this exact definition in the calculation of precision and recall of the algorithm on the out-of-sample results. The results of the performance test in the identification task are shown in the Supplementary Information section A.4.

Regarding the green-washing cases, once a company reports an initiative as an implemented action, and the algorithm correctly identifies it, we do not verify, ex-post, the true value of the statement. This is a limitation of our approach and we agree with you that it deserves a discussion, which you can now find in the limitation section (see lines 472-480). Thank you for the suggestion. We have also augmented the model to address the possible impact of green-washing initiatives on our results by controlling, in every regression specification, for whether a report was subject to an external audit process (see lines 695-710 for more detail on how we introduce this control factor in the model). Auditing should limit the frequency of green-washing because statements are verified by third parties. Therefore this control factor helps us to isolate the effect of green-washing in our estimations. However, green-washing remains a very important issue to investigate in further depth in future works. This is surely on top of our minds. We have made this point clear in lines 472-482 of the new version of the manuscript.

• *On page 11, the authors mention that they “control for self-selectivity on the disclosure of the emissions” without specifying how this happens (unless I have overlooked something).*

Answer. Sorry for the lack of clarity. We have now dedicated an entire section in the Material and Methods to describe the model in greater detail (including our approach to account for self-selectivity, see section "Sustainability initiatives and alignment with climate targets" in the new version of the manuscript on page 26). Specifically, because companies that issue sustainability reports might differ systematically from non-publishing companies, but we only observed the reports (and the behaviour) for the companies that issue one, there is the potential for self-selectivity in our sample. To address this source of endogeneity, we estimate the model using the Heckman correction. First, we run a Probit model where the independent variable is one if a company issues a report in year Y and zero otherwise. Then we use the inverse Mills ratio from this step in the linear model that explains the magnitude of the deviation from the target. The coefficients of the first step Probit regression are shown in table ST7 of the new version of the manuscript.

For what concern the emissions, we take a different approach. Because Trucost also estimates the emissions of companies that do not disclose them in detail (using the GEVA Modelled approach, as described in section "Climate targets" on page 25 of the new version of the manuscript), we cannot correct for the self-selectivity of emission disclosure with the Heckman correction (i.e., because we also observe data for companies that did not disclose enough emission data). Instead, in our approach

we have identified the data source from Trucost and distinguished between observations with estimated emissions and observations with disclosed emissions (this information is provided directly from Trucost in their data feed). Then we added an indicator variable in the regression model that takes the value of one if Trucost uses disclosed emissions and a value of zero if they estimate them. Thank you for pointing out that this section needed more clarification. We have now significantly expanded the section in the Material and Methods on the model specification to help the reader understand our estimation approach, see page 26, section "Sustainability initiatives and alignment with climate targets" in the new version of the manuscript.

• *Third, the authors include companies from different countries in the dataset which operate within different national legal frameworks when it comes to nonfinancial reporting. For instance, during the time period considered by the study, the nonfinancial reporting directive (NFRD, Directive 2014/95/EU) in the European Union has made it mandatory for the large companies with over 500 employees to report on environmental, social and governance matters. Hence, the criteria for nonfinancial reporting are different than for other companies where such regulation does not apply. The authors might consider an additional type of analysis to understand if the results are qualitatively similar in mandatory vs. voluntary sustainability reporting settings.*

Answer. Point well taken. We are now controlling for country-level fixed effect as opposed to geography fixed effect. The difference is not as relevant for countries within the European Union (which follow the same reporting regulations). However, it is essential for countries in other geographical regions such as, for example, the Asia-Pacific, which includes companies in countries as diverse, from a regulatory perspective, as Australia, China, and Japan, to cite a few. Notice that the estimation process only covers the observation period from 2018 to 2020 and uses information dating back to 2016. Therefore, the observation period for estimating the effect does not include substantial regulatory changes such as the introduction of the European NFRD, the Australian NGER and the US GHGRP. Importantly, we have included country fixed-effects in each and every specification in the new version of the manuscript. Note as well that country-specific controls enter in the regression through the inverse Mills ratio, which is derived by also controlling for the probability of a company in a given country issuing a sustainability report in the first step of the Heckman correction (see lines 717-730 in the new version of the manuscript). This probability is directly related to the mandatory vs. voluntary reporting setting of the given country.

Reviewer 3

• *This paper develops a novel proxy for corporate initiatives based on automated textual analysis of corporate sustainability reports. The authors focus on correlations between their novel proxy and corporate emissions, with a focus on alignment with Paris Agreement emissions trajectories. Emissions data comes from Trucost, a third-party data vendor. The main results are twofold: first, the total number of initiatives is not significantly associated with the level of emissions or Paris alignment. Second, and perhaps more interesting, certain types of initiatives are positively associated with Paris Alignment, while others are negatively associated. The authors frame their contribution in terms of developing a measure and framework that can help guide capital allocation and policy.*

Answer. Thank you for taking the time to review our manuscript and for the thoughtful comments and suggestions. Below we respond to your comments and point to new discussions and analyses in the new version of the manuscript, which we believe address your concerns.

• *1. Measurement: Unfortunately, there are important unaddressed issues in both the primary independent and dependent variables.*
a. Sustainability reports: These are voluntary disclosures, such that firms have significant latitude in what they contain. Often, these are glossy, almost marketing documents. Firms also have significant discretion in how they portray any initiative. Furthermore, in measuring initiatives, I was surprised that the authors chose to only look at three consecutive sentences. Often, firms will devote entire pages to initiatives, including pictures and even plots. Overall, I am concerned that the authors have not sufficiently considered how the discretionary nature of the content of sustainability reports might affect the properties of measured initiatives, and how that might filter into associations with emissions and Paris Alignment.

Answer. Thank you for pointing out this important limitation that we are now addressing by refining the specification of our regressions. Regarding the length of the initiatives. It is true that some initiatives are described at length in the reports. However, in our experience, from extensive reading of sustainability documents, most of the initiatives are described in short paragraphs, and most of the content is included within two or three sentences (e.g., see Table ST2 as an illustrative example). Indeed, this is also the case for initiatives reported in other sustainability documents, such as the Carbon Disclosure Projects (CDP). However, we agree with your point that further care was needed to control for the characteristics of the reports that could influence the properties of measured initiatives. We, therefore, performed additional analyses to better account for these factors. Specifically, we have implemented the following changes:

1. Part of the discretionary nature of the reporting can be directly related to whether or not disclosing nonfinancial information is regulated at the country level. In the previous version of the manuscript, we controlled for differences in voluntary versus mandatory reporting frameworks, which influence the quality and scope of the reports, by including geographies fixed effects. However, we are now controlling for countries, as opposed to geography, fixed effects in each regression specification. Country fixed effects are more accurate controls because while in certain regions, such as Europe, regulatory frameworks on non-financial disclosure across countries are comparable (due, for example, to the

NFRD, Directive 2014/95/EU), in other regions, such as the Asia-Pacific, there can be substantial differences that require country-specific controls;

2. Note as well that country-specific controls enter in the regression through the inverse Mills ratio, which is derived in the first stage of the Heckman correction by also controlling for the probability of a company in a given country issuing a sustainability report.
3. We are now also controlling for the characteristics of the reports (see the answer to your next comment for a more detailed discussion on this control variable). In particular, we control for whether a company follows GRI guidelines in their reporting and if the report has gone through an external audit process. Controlling for these factors allows us to account for differences in the content of the reports. Indeed, following reporting standards and having the report verified by third parties can have a substantial impact on what and how companies disclose in their reports.

Overall, these new control factors helped us differentiate reports subject to different scrutiny, which is a factor related to the properties of the measured initiatives. We have implemented these changes in every specification in the new version of the manuscript and described them in section "Sustainability initiatives and alignment with climate targets" on page 26.

• *I was also surprised by statistics on the sustainability reports. They seem to diverge quite a bit from those provided by Gipper et al. (https://papers.ssrn.com/sol3/papers.cfm?abstract_id=4263085).*

Answer. As you correctly pointed out, our data here differ from those reported in other studies, such as the one you referenced. We have investigated the reason for these differences and found the origin of the discrepancy. To characterise the reports, we use data from Refinitiv Asset4, which is by far and large the leading data provider of ESG and nonfinancial data for market participants and academics (see, for example *de Villiers, C. Accounting & Finance, 62, 4523–4568* which is a critical evaluation of the use of Refinitiv Asset4 in the corporate sustainability literature, and *Ioannou I. and Serafeim Journal of International Business Studies. 43, 834–864*, which also describes Asset4 in great depth.). For each company-year with a published sustainability report, Asset4 assigns a 1/True (0/False) if the report display (does not display) a particular characteristic. To be concrete, if, for example, company X in year Y follows GRI guidelines in their sustainability report, Asset4 assigns a 1/True to that observation. However, the Asset4 database contains several missing data. Missing data occur when Asset4 analysts are not able to conclusively assert if the report display or does not display a specific characteristic (this was confirmed to us by Asset4 directly). This ambiguity could impact the result of our analysis. Therefore, we run an additional test to assess the robustness of the result due to this possible source of error in the control variables.

Specifically, we run two specifications of the models including control variables for the characteristics of the report. In the first specification (A), we consider missing values as a separate class and add fixed effects for 0/False, 1/True, and missing values. In the second specification (B), we consider missing values as zeros, and we use

an indicator for the presence or absence of the characteristic. Because the behaviour is estimated on a rolling basis in the models, we include the fixed effects and the indicator variable for every year in the estimation window. Notice that when treating missing values as zeros, our numbers match quite closely the numbers reported in the referenced paper. We have illustrated this point by also re-drawing panel **d** in Figure S4 and S5 in the new version of the manuscript, including two lines for each characteristic: one for the data used in (A) and one for the data used in (B). However, because it is unclear which approach we should be following (dropping missing values or treating them as zeros), we have performed the two different analyses and reported the results of each of them. In the main text, we report the results of specification (A) because it involves minimal data interpretation of the Asset4 dataset. In the Supplementary Information Table ST9 we report the result of specification (B). The two sets of coefficients are qualitatively similar. We have now discussed these important points in lines 695-710 of the new version of the manuscript.

• *Emissions: Trucost sources emissions information from firm disclosures and proprietary estimates. If coming from firm disclosures, the authors should consider how colocation in the same document as the initiatives might affect associations. If coming from estimates, the authors need to consider what is driving the firm to provide insufficient emissions disclosures such that Trucost needs to estimate emissions. Additionally, Trucost estimates are likely to be noisier than firm-derived emissions measures, which could systematically bias the analysis of associations between initiatives and emissions. Specifically, firms that provide less information about initiatives are also likely to provide less information about emissions, though there may be other selection mechanisms at play.*

Answer. Point well taken. We have now collected information from Trucost to distinguish between observations with emissions reported from corporate disclosure and emissions estimated by Trucost from their internal models (Trucost provide this information in their data feed). Then we re-estimated our models controlling for a binary indicator that takes the value of one if the value is estimated from a reported emission data point and zero if Trucost estimated the underlying emissions data. We now include this control factor in every specification in the new version of the manuscript. We have explained our approach in detail in section "Sustainability initiatives and alignment with climate targets" on page 26 of the new version of the manuscript. We have also provided more detail on Trucost methodologies and how they deal with missing data in section "Climate targets" on page 25 in the new version of the manuscript.

• *2. Causality and scope: The authors are clear that readers should not infer a causal relation from their analysis. However, the causal relation is presumably the most important part. I was left curious about whether the initiatives facilitated alignment or if firms with aligned objectives chose to engage in systematically different types of initiatives. I am not sure that showing a correlation between types of initiatives and Paris Alignment provides a sufficient contribution. A high-level contribution might come from showing what types of initiatives drive emissions decreases that facilitate alignment, but that does not seem to be feasible from the current setting.*

Answer. Sorry for the lack of clarity. In our empirical approach, we address the problem of reverse causality by estimating the regression specifications using historical averages of behaviour and asset characteristics (the independent variables), see

lines 679-685. Using historical averages for the independent variables reduces the possibility that our results suffer from reverse causality problems because we do not expect that alignment calculated in a given year can influence average behaviours (i.e., the choice to engage in systematically different initiatives) during the previous years.

Regarding the correlation versus causation problem, because Nature Communications is a general audience journal, we believe that it is extremely important that non-technical as well as technical readers correctly understand our results and their implications. This is particularly relevant for readers that are not familiar with the nuances of empirical research. Therefore, we have dedicated an important discussion to this topic (see lines 450-471 in the new version of the manuscript). First, no empirical work can conclusively claim causal explanations. In this work, we have followed standard practices and collected a substantial amount of data to control for all the factors that could potentially bias our estimates. Specifically:

- We control for asset characteristics that can be associated with both behavioural choices and alignment (I.e., potential confounding factors)
- We control for self-selectivity in the disclosure of nonfinancial information
- We control for systematic differences at the sector and country levels through fixed effects
- We control for differences in the characteristics of the reports
- We control for differences in the sources of emission data
- We repeat the estimation across different sub-sets of our sample, I.e., across size quartiles and years of estimation of the alignment with the climate targets.

However, because our work is empirical, we still cannot claim causality. To be specific, our evidence points towards a statistically significant association between a given set of sustainability activities and goals (i.e., sustainability behaviour) and alignment with climate targets. There are two possible interpretations for these associations. In the first interpretation, the associations imply a direct (causal) link between initiatives and alignment. Indeed, most of the initiatives reported in the analysis can be directly related to emission reduction processes. For example, we expect that developing new products and investing in research and development for developing and employing renewable energy sources will result in lower future emissions. Interestingly, however, as shown in Table ST15, non-causative initiatives, such as donation & funding, also seem to play a role, albeit small. Therefore, our result can also imply the existence of a more complex latent causal structure. For example, companies that donate substantial capital to emission reduction activities might be more committed to sustainability issues and adopt behaviours that result in lower long-term emissions. In the second interpretation, our dataset provides a window to these latent causal structures (behavioural choices or sustainability strategies) that we do not explicitly infer here.

Following our discussion, as you correctly suggest showing what types of initiatives drive emissions decreases that facilitate alignment is not feasible from the current

setting because such a high-level analysis would require the development of field experiments. We believe that this could be a very exciting future avenue of research that would require a substantial level of funding and the creation of long-term collaborations with businesses. Our analysis could potentially form the basis for the design of these type of experiments.

REVIEWERS' COMMENTS

Reviewer #1 (Remarks to the Author):

Thank you for providing clarifications in response to my questions 1 and 2. I find your explanations satisfactory for these points. However, regarding my third question, I remain unconvinced about the overall contribution of this paper. A deeper analysis of the corporations' hazard-related actions and associated risk categories, in relation to their environmental impact, seems absent. It is essential to conduct an in-depth analysis of different risk categories and assign them varying weights in the final formula. Assuming equal weights for all risk categories may not offer an accurate reflection of their actual environmental implications.

Reviewer #2 (Remarks to the Author):

I have read the revised version of the manuscript and the authors' response to the reviewers' comments. I feel that the authors have done an excellent job in addressing my concerns and suggest no further changes.

Response to Reviewers: The alignment of companies' sustainability behavior and emissions with global climate targets

Simone Cenci¹, Matteo Burato¹, Marek Rei^{1,2} and Maurizio Zollo¹

¹Leonardo Centre on Business for Society, Imperial College Business School, London, UK

²Department of Computing, Imperial College London, London, UK

Reviewer 1

• Thank you for providing clarifications in response to my questions 1 and 2. I find your explanations satisfactory for these points. However, regarding my third question, I remain unconvinced about the overall contribution of this paper. A deeper analysis of the corporations' hazard-related actions and associated risk categories, in relation to their environmental impact, seems absent. It is essential to conduct an in-depth analysis of different risk categories and assign them varying weights in the final formula. Assuming equal weights for all risk categories may not offer an accurate reflection of their actual environmental implications.

Answer. Thank you for the positive feedback. Adding different weights to the final model would force us to include an arbitrary choice in the analysis, which we prefer to avoid. Our agnostic, data-driven, approach provides a robust baseline for analysing the association between different behavioural choices and alignment of corporate emissions with global climate targets. This is the main rationale behind our idea of performing an ex-post analysis, as we now discuss in line 309 of the new version of the manuscript.

Reviewer 2

- *I have read the revised version of the manuscript and the authors' response to the reviewers' comments. I feel that the authors have done an excellent job in addressing my concerns and suggest no further changes.*

Answer. Thank you for taking the time to review our manuscript again. We also would like to take this occasion to thank you again for your feedback.